# Asymmetric activity of NetrinB controls laterality of the *Drosophila* brain

F. Lapraz [1] ✉, C. Boutres[1], C. Fixary-Schuster [1], B. R. De Queiroz[1], P. Y. Plaçais [2], D. Cerezo [1], F. Besse [1], T. Préat [2] & S. Noselli [1] ✉

Left-Right (LR) asymmetry of the nervous system is widespread across animals and is thought to be important for cognition and behaviour. But in contrast to visceral organ asymmetry, the genetic basis and function of brain laterality remain only poorly characterized. In this study, we performed RNAi screening to identify genes controlling brain asymmetry in *Drosophila*. We found that the conserved NetrinB (NetB) pathway is required for a small group of bilateral neurons to project asymmetrically into a pair of neuropils (Asymmetrical Bodies, AB) in the central brain in both sexes. While neurons project uni-laterally into the right AB in wild-type flies, *netB* mutants show a bilateral projection phenotype and hence lose asymmetry. Developmental time course analysis reveals an initially bilateral connectivity, eventually resolving into a right asymmetrical circuit during metamorphosis, with the NetB pathway being required just prior symmetry breaking. We show using unilateral clonal analysis that *netB* activity is required specifically on the right side for neurons to innervate the right AB. We finally show that loss of NetB pathway activity leads to specific alteration of long-term memory, providing a functional link between asymmetrical circuitry determined by NetB and animal cognitive functions.

Asymmetry is a fundamental feature of biological systems, playing essential roles in cell, tissue as well as whole-body organization and function. Left-Right (LR) asymmetry of the nervous system is wide-spread in bilaterian[1,2], occurring at all scales (organ morphology, neuron number or activity, connectivity, and gene expression) but also in the form of lateralized behaviours, familiar examples in human being left hemisphere processing of language[3] and handedness[4]. Meta-analysis from thousands of MRI scans showed a wide array of structural asymmetries (e.g., cortical thickness) in the human brain[5], and defec-tive hemispheric lateralization has been associated with several cog-nitive disorders, including Dyslexia, Schizophrenia, attention-deficit/hyperactive and autism spectrum disorders[6-9]. Work in chicken, worm and fish[10] have revealed a role for external environmental clues (e.g., light), and intrinsic developmental programs (e.g., nodal) in nervous system laterality. Orientation of the chick embryo in the egg is such that the right eye is exposed to light while the left one is occulted. Light

exposure of the right eye for as little as 2 h during day 19 of develop-ment is sufficient to elicit lateralized attack and copulation responses[11]. In addition, unilateral exposure of the right eye induces asymmetry in fibres connecting the visual thalamus[12], which is associated with the specialized use of right and left eyes for discriminating grains from pebbles and detecting predators, respectively[13]. But lateralized function of the brain is not always associated with anatomical LR dif-ferences. In the nematode, a pair of bilaterally symmetrical neurons (ASER and ASEL) involved in sensing taste (soluble salts) can only be discriminated through the differential expression of chemoreceptors[14]. In vertebrates, one of the most studied asymme-trical brain structures is the habenula, a conserved bilateral region of the epithalamus involved in response to fear and conditioned aversive stimuli. Habenula nuclei show lateralized morphology and neuronal projections[15], particularly in the zebrafish, where parapineal gland precursors show directional migration towards the left side, a

[1]Université Côte d'Azur, CNRS, Inserm, iBV, Nice, France. [2]Plasticité du Cerveau, UMR 8249, CNRS, ESPCI Paris, PSL Research University, Paris, France. ✉e-mail: francois.lapraz@unice.fr; noselli@unice.fr

developmental process relying on conserved FGF8 and nodal signalling pathways[10]. The parapineal gland sends projections specifically in the left habenulae, and left and right habenula receive afferent visual and olfactory inputs, respectively, hence connecting laterality with animal behaviour. Yet, despite brain LR asymmetry being prevalent, genes and mechanisms controlling brain laterality establishment, development, and function remain poorly characterized.

In this work, we reveal the conserved *Drosophila* NetrinB pathway as being required to build an asymmetrical circuit, the H-neurons, in the central complex of the brain. Using unilateral clonal analysis, we further show that netB ligand has a unilateral activity in the right side of the brain. Finally using two different and complementary paradigms, aversive memory assay and courtship memory assay, we show that asymmetrical H-neurons are necessary for long-term memory of adult flies.

## Results

### The *Drosophila* 'H-neurons' as a model to study brain lateralization

To better understand the origin and function of brain laterality, we set to develop *Drosophila* as a novel, genetically amenable model system. The *Drosophila* central nervous system is essentially symmetrical, except for a bilateral structure known as the asymmetrical bodies (AB)[16,17]. The AB form a pair of neuropils, hereafter referred to as left AB (LAB) and right AB (RAB), present on each side of the midline in the central complex (Fig. 1a), a major structure of the adult brain integrating spatial control and motor activity[18–20]. Interestingly, a subset of bilateral neurons belonging to the LALv1A lineage[21] project axons asymmetrically into the RAB only[16,17]. These neurons, hereafter termed 'H-neurons' because of the overall shape of the circuit, can be visualized using the *72A10* driver lines[22]. H-neurons show a clear asymmetric innervation pattern towards the RAB in ~95% of wild-type flies (hereafter referred to as ASYM flies) (Fig. 1a–d), with the remaining ~5% of the population showing spontaneous bilateral innervation (referred to as SYM flies) (Fig. 1a, e–g).

To further characterize the asymmetry of AB and H-neurons, we first performed a quantitative 3D-analysis of their organization and morphology. In ASYM flies, the RAB is 4-times larger than the LAB, with SYM flies showing a LAB/RAB ratio of 0,59[17] (Fig. S1a). Morphometric analysis indicates that ASYM and SYM flies have the same number of H-neurons on each side (9 on average) and do not differ in several metrics (neurite length, landmark spacing)(Fig. 1h, i; Fig. S1b, c), indicating that a difference in cell number does not account for the observed asymmetry of the circuit and that neurons in ASYM/SYM flies show equivalent proportions. We also found that apoptosis does not play any role in the asymmetry of the H-pattern (Fig. S2a). To determine the basis for neuronal asymmetry, we next characterized the morphology of individual H-neurons by single neuron labelling using the MCFO technique[23] (see Methods). The ASYM/SYM phenotype of flies was determined with the 72A10-LexA driver (to reveal H-neurons) and Fas2 staining (specifically marking the RAB neuropil). For clarity, we describe H-neuron morphology as follows: a prefix letter (L or R) refers to the position of the cell body in the Left or Right hemisphere, followed by a second term indicating axon projection into LAB, RAB or both Left and Right AB (LRAB). Accordingly, the 6 possible neuron trajectories are L-LAB, L-RAB, L-LRAB and R-LAB, R-RAB, R-LRAB (Fig. 1j–m). Results show that ASYM flies only have two types of neurons, L-RAB and R-RAB, indicating that both left and right H-neurons project into the RAB through contralateral and ipsilateral innervation, respectively (Fig. 1j, k). In contrast, SYM flies have their left and right neurons projecting into both AB (L-LRAB and R-LRAB) (Fig. 1l, m), well consistent with their bilateral phenotype. We could observe an instance of a RAB-only projecting neuron (L-RAB) (Fig. 1l, right panel), which may explain why SYM flies, although showing clear bilateral innervation of the ABs, still retain a moderate bias towards right (LAB/RAB

ratio <1; Fig. S1a). Our analysis helps to rule out some alternative possibilities, including an ipsilateral connectivity scenario, in which left neurons would specifically project into the left AB while right neurons would project into the right AB. The L-RAB, L-LRAB, R-RAB, R-LRAB neurons found in wild-type ASYM and SYM flies have a generic name, SLP-AB[17], for which we determined their LR features, including the position of their cell body and their specific distribution into SYM and ASYM flies. We analysed the recently published hemibrain connectome indicating that the female used was an ASYM fly, with 8 L-RAB neurons on the left and 9 R-RAB neurons on the right (Fig. S6), in agreement with our data (Fig. S1b).

To understand the developmental origin of H-neuron asymmetry, we performed time-course analysis during pupal development (Fig. 2a–e). Data show that H-neuron innervation is first visible at 28 h After Puparium Formation (APF), as an emerging bilateral pattern, persisting till 32 h APF. At 34 h APF, asymmetry starts to be visible with the RAB showing a stronger signal and larger volume (Fig. 2b–d). From 34 h to 48 h, the pattern gradually resolves into a fully asymmetrical circuit, following a linear growth (blue curve in Fig. 2b). During the formation of the pattern, we did not observe any difference in the number of cell bodies or morphometry (Figs. 2e and S3 left panel), however asymmetry in AB's volume started to emerge at 32 h APF and increased steadily leading to the pervasiveness of the ASYM phenotype (Fig. 2d; Fig. S3 right panel). These results indicate that bilaterality corresponds to the default state with symmetry breaking taking place at 32 h/34 h APF during metamorphosis (Fig. 2c).

### The NetrinB pathway controls H-neuron asymmetry

To unravel the mechanisms underlying *Drosophila* brain asymmetry, we first tested the role of the conserved *myosin1D* (*myo1D*) gene[24,25], which controls LR asymmetry of all the visceral organs[26–37]. We observed that a *myo1D* null mutant did not show any phenotype in AB or H-neuron asymmetry, indicating that brain and visceral laterality are controlled by distinct mechanisms (Fig. S2b). Since the stereotyped routing of H-neurons towards the RAB suggests a role for axonal guidance and midline crossing (left and right H-neurons show different midline crossing behaviours), we performed an RNAi screen targeting factors involved in motor axon guidance (axons repelled by midline)[38] and midline crossing (axons attracted by midline)[39], representing a group of 61 candidate genes (Fig. 3a; Data S1). Two RNAi lines for each gene were expressed in H-neurons with the *72A10-Gal4* line and screened for defects in their AB connectivity. Of 61 candidate genes, 10 showed significant loss of H-circuit asymmetry (Fig. 3a, b). Driving RNAi against the 10 positive hits using a pan-glial driver (*gcm-Gal4*) did not show a phenotype, indicating that the laterality defects are neuronal-specific (Fig. S2c).

In this study, we focus on the role of the Netrin (Net) pathway. Indeed, gene network reconstruction from positive hits showed a prevalence for Net pathway genes, including the two receptors (Frazzled[40] and Unc-5[41]), as well as known transducers (RhoGEF64C[42], mud[43], Pak[44], and Src64B[45])(Fig. 3b, c). Of note, *unc-5* silencing showed the strongest defect with a fully penetrant SYM phenotype (Fig. 3b). Because silencing of Netrin ligands (NetA and NetB) did not show a phenotype using the restricted 72A10-Gal4 driver, we tested their potential role using a range of Gal4 lines. Interestingly, expression of RNAi against NetB using either *elav-Gal4* or *period-Gal4* (which includes the H-neuron lineage) led to a fully penetrant bilateral phenotype, indicating that NetB is necessary for the process but not required in the H-neuron circuit itself (Fig. 3d, e). Importantly, we further show that *unc-5* and *netB* null mutations, despite being viable, led to a strong bilateral phenotype, thus confirming our RNAi results (Fig. 3d, e). Of note, *NetA* loss-of-function (either using RNAi or a *NetA* null mutant) did not show any phenotype, indicating NetB specificity in H-neuron asymmetry (Fig. 3d, e).

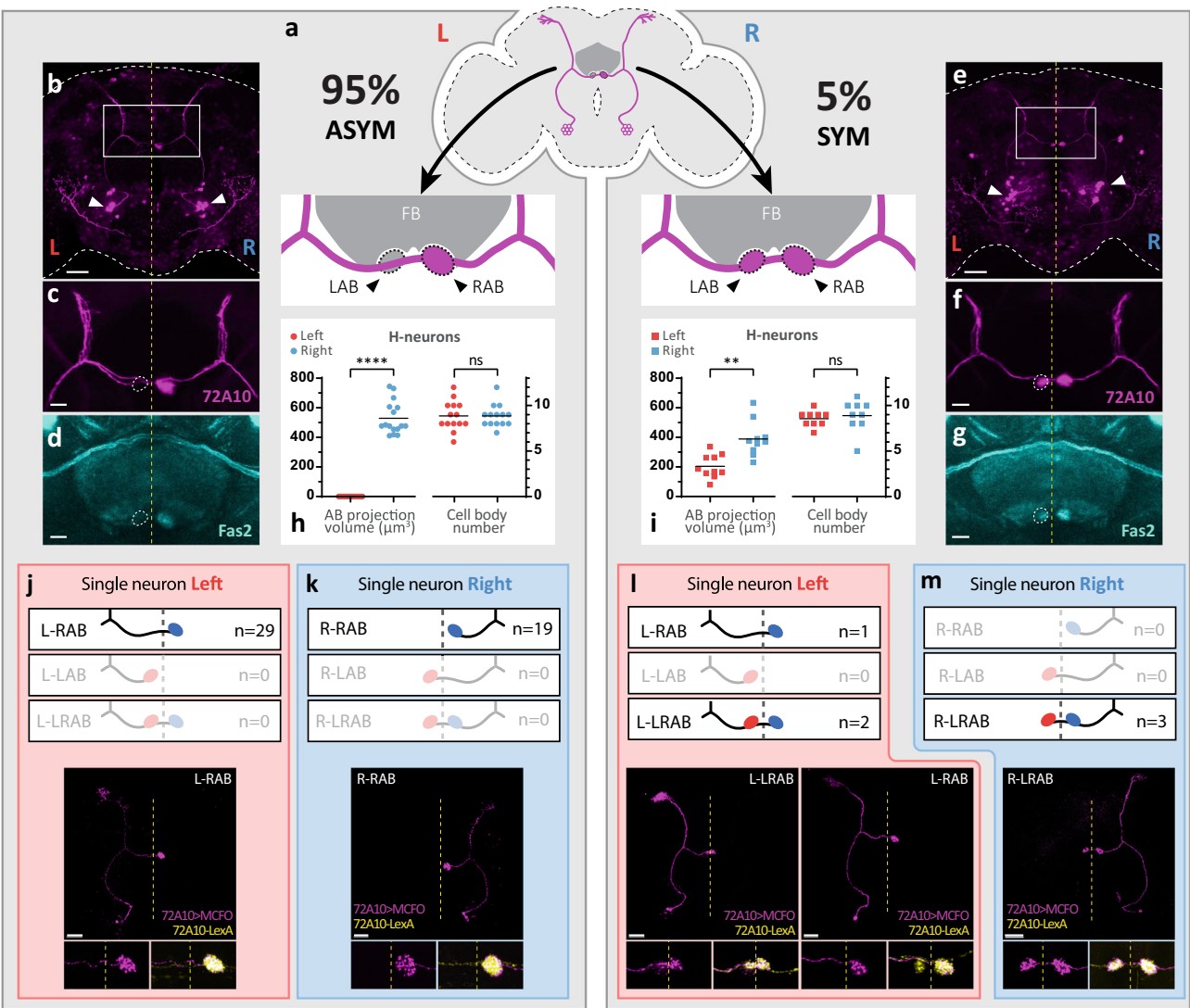

**Fig. 1 | Characterization and development of asymmetric H-neurons.**
**a** Stereotyped asymmetry of AB-projecting H-neurons, marked using *72A10-lexA* driver. Two categories of flies are found in wild-type: 95% of flies (ASYM flies; left panel) have H-neurons projecting asymmetrically into the Right AB (RAB), while in the remaining 5% flies (SYM flies; right panel) projections are symmetrical and found in both Left and Right ABs (LAB and RAB) (*n* = 102 brains). **b**–**d**, **e**–**g** On average, 9 H-neurons (cell bodies shown as white arrows in **b** and **e**) labelled with *72A10-LexA* driver (magenta) are found per hemisphere. H-neurons's AB projections show Fas2 immunoreactivity (cyan) in both ASYM **d** and SYM flies **g**. Dotted line (**c**, **d**, **f**, **g**) represent LAB position. Scale bars represent 30 μm **b**, **e** and 10 μm **c**, **d**, **f**. **g. h**, **i** The volume of the AB shows clear asymmetries within and between each category of flies (ASYM flies, left and right: *n* = 16; SYM flies, left and right: *n* = 10). However, there is no asymmetry in the number of H-neuron cell bodies in either ASYM **h** or SYM flies **i** (ASYM flies, left and right: *n* = 14; SYM flies, left and

right: *n* = 9). *n* numbers correspond to the numbers of hemispheres analysed. Horizontal bars represent mean value. Two-tailed Wilcoxon-test p-Values are: ASYM AB volume: <0.0001; ASYM cell body number: >0.9999; SYM AB volume: 0.0098; SYM cell body number: 0.6719. Significance threshold for *p*-Value are: ns, non-significant; *<0.05; **<0.01; ***<0.001; ****<0.0001. Source data are provided as a Source Data file. **j**–**m** Single-cell clone analysis of H-neurons. All 6 possible neuron trajectories (L-LAB, L-RAB, L-LRAB, R-LAB, R-RAB, and R-LRAB; see text for details) are schematized and their occurrence in ASYM **j**, **k** and SYM **l**, **m** flies indicated as n numbers. Single neurons are labelled with the *72A10-Gal4* driver through the MCFO method (magenta) while the whole H-neuron circuit is labelled with the *72A10-lexA* driver (yellow), allowing to determine whether an ASYM or SYM fly is observed. Insets at the bottom show higher magnification of AB region with representative single neuron types (magenta) and the whole H-neuron circuit (yellow). Dotted line represents the brain midline. Scale bars, 30 μm.

We next characterized NetB pathway loss-of-function phenotypes. H-neuron cell body counting, and morphometric analysis indicate that the *netB* mutant and *unc-5 RNAi* phenotype is similar to spontaneous wild-type SYM flies (Fig. S1). We noted that the AB volume of *netB* mutants is increased as compared to SYM flies (Fig. S1a), which is not linked to H-neuron number, and may be related to additional, unspecific effects of the mutation. The characterization of single neuron morphology of *NetB* mutants and *unc-5-RNAi* flies shows that, like wild-type SYM flies, the bilateral phenotype is due to the presence of bilateral neurons (L-LRAB, R-LRAB) with some rare unilateral ones in the *netB* mutant (Fig. 4a; Fig. S7). The

*unc-5* RNAi led to more unilateral neurons (L-RAB, R-RAB) (Fig. 4b; Fig. S7), due to a hypomorphic effect of RNAi silencing in these conditions (Fig. S1a). Hence, *netB* mutant flies lack asymmetry and are no longer capable of making right-only innervation, suggesting that in these flies symmetry breaking does not occur. We confirmed this hypothesis by performing time course analysis during metamorphosis, showing that in *netB* mutants, H-neurons do not resolve into an asymmetrical circuit and remain in the default, bilateral phenotype (Fig. 4c–g). Comparison of wildtype (Fig. 2b) and *netB* mutant (Fig. 4d) curves suggests the presence of two phases: one *netB*-independent phase (28–42 h) during which bilateral rightward

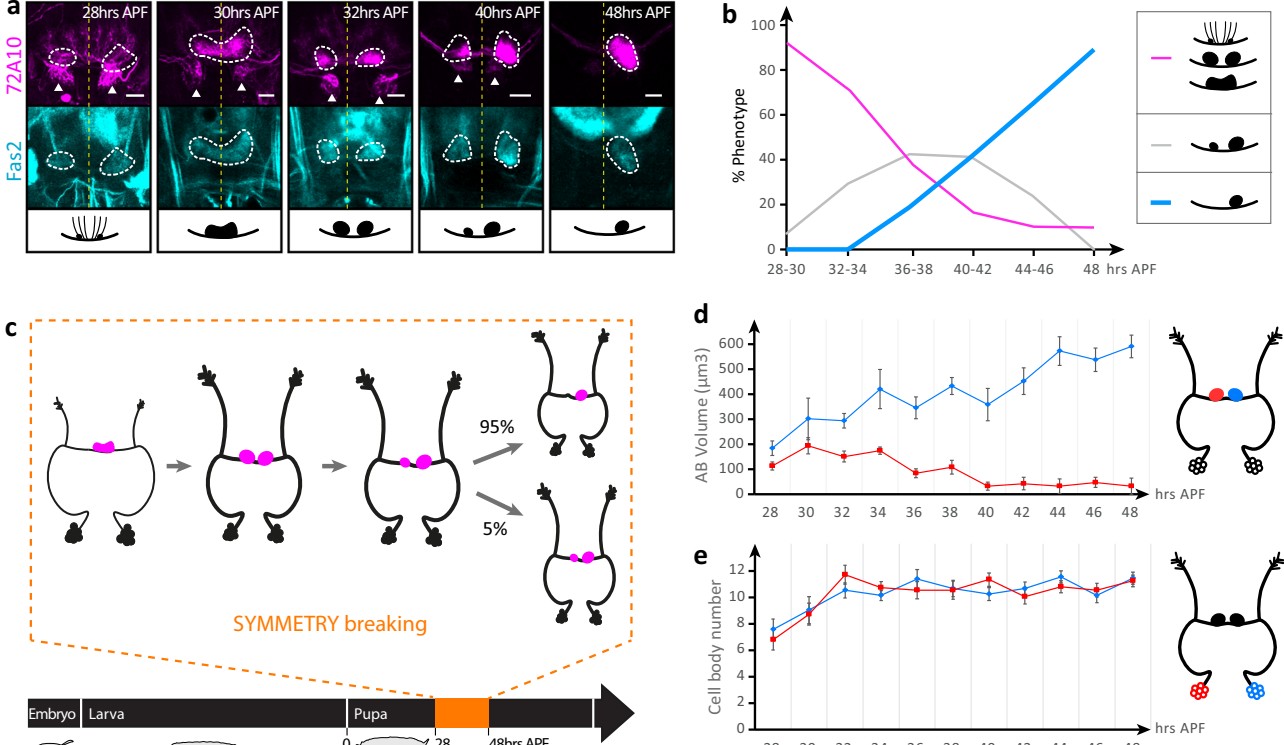

**Fig. 2 | Development of the H-circuit during pupal stages. a** H-neuron projections are initially symmetrical (28 h APF) and resolve into an asymmetrical circuit (48 h APF) during an ~20 h-long process. Confocal images of representative H-neurons (magenta) stained with anti-Fas2 antibody (cyan) during pupal stages. ABs (white dotted lines) and noduli (white arrows) are shown. Projections of H-neurons in the different categories are as follows (from left to right): "AB nucleation", vertical projection in fan-shaped body (FB), in noduli and symmetrical in left and right AB primordia; "Bilateral fused", symmetrical projection in fused left and right ABs; "bilateral", symmetrical projection in separated left and right ABs; "Bilateral with right bias", asymmetrical projection in separated left and right AB (right AB volume >3× left AB volume); and "Right", asymmetrical projection in right

AB exclusively. Scale bars, 10 μm. **b** Evolution of the H-circuit during pupal development (28–48 h APF) showing a steady state increase of the ASYM phenotype. (*n* = 20 brains for 28–30, 32–34, 36–38, 40–42, and 44–46 h APF representing the merging of 2 adjacent data points shown in d; *n* = 10 for 48 h APF). Source data are provided as a Source Data file. **c** Summary of the emergence of H-neuron asymmetry during pupal stage. **d**, **e** Morphometric analysis of H-neurons during pupal development (28–48 h APF), showing the evolution of the volume of left and right AB projections **e**, and the number of left and right H-neuron cell bodies **d**. *n* = 10 brains per time point. Data are presented as mean ± standard error of the mean (SEM). Source data are provided as a Source Data file.

biased brains increase and reach a plateau or maximum (grey curves in Figs. 2b and 4d); and second, a *netB*-dependent phase during which bilateral rightward biased brains decrease (40–42 h) for the benefit of right-only or bilateral circuits, as is observed in wildtype and *netB* mutants, respectively.

To understand how NetB could control laterality, we first determined the functional time window for *netB* and *unc-5* genes in H-neuron asymmetry, using a temperature-dependent RNAi ON-OFF method (see Methods). Results show that *netB* expression is required at late larval stage preceding puparium formation (Fig. 4h). *unc-5* activity in turn is required later in development, at early pupal stage prior to symmetry breaking, consistent with a ligand-receptor model of activation, as observed in mice[46,47]. Next, we analysed the expression pattern of *netB* and *unc-5* at different developmental stages using a series of reporter lines. NetB expression could not be determined due to diffuse, unspecific pattern in the H-neurons and/or AB. Additionally, we found that commonly used *netB* reporters, including NetB::GFP[48–50], NetB-TM, NetB-myc[51] and NetB-TJ-Gal4[52], induced a bilateral phenotype on their own (Fig. S2d), making them unsuitable in the context of brain laterality. However, we could use a Unc-5::GFP reporter that do not induce an asymmetry phenotype to detect the Unc-5 protein in the AB (Fig. S4). Unc-5::GFP expression is first bilateral at stage 24 h APF and then resolves into an asymmetrical signal in the RAB at 48 h APF, mirroring the emergence of the 72A10-lexA pattern during pupal development (compare with Fig. 2a).

## Unilateral activity of NetrinB controls H-neuron laterality and long-term memory

How is the NetB pathway controlling laterality? An instructive model for NetB activity predicts that the pathway would be active only on one side of the brain (left or right). To test this hypothesis, we performed unilateral knock-down of NetB pathway genes by carrying out low frequency clonal analysis (see Methods). Generating single clones from the *period-Gal4* driver allowed us to produce *netB-RNAi* or *unc-5-RNAi* left or right clones in 9 specific groups of neurons altogether composing the whole *period-Gal4* pattern. The 9 groups include 4 known lineages (ALv1, EBa1, LALv1A, BAlp2), 3 lineage subsets (SLPpl1, DPLm2, LG-N neurons) and 2 other neuronal groups (hereafter referred to as Ventral1 and Ventral2, whose morphology does not match any previously reported neuronal lineage)(Fig. 5a; see Methods). Analysis of 84 single and 139 multiple clones shows that *unc-5* is required in the LALv1A lineage (to which the H-neurons belong), both in left and right neurons (Fig. 5a, b), well consistent with our genetic and expression data. Interestingly, analysis of 102 single and 171 multiple clones indicates that NetB is required in the same LALv1A lineage, but only on the right side (Fig. 5a, c). Hence, the NetB ligand shows unilateral activity, which is required to establish the asymmetric H-pattern in an *unc-5*-dependent manner. The fact that silencing *netB* using the *72A10-GAL4* driver does not induce a phenotype (Fig. 3b, c) indicates that NetB activity resides in neurons belonging to the right LALv1A lineage, excluding the

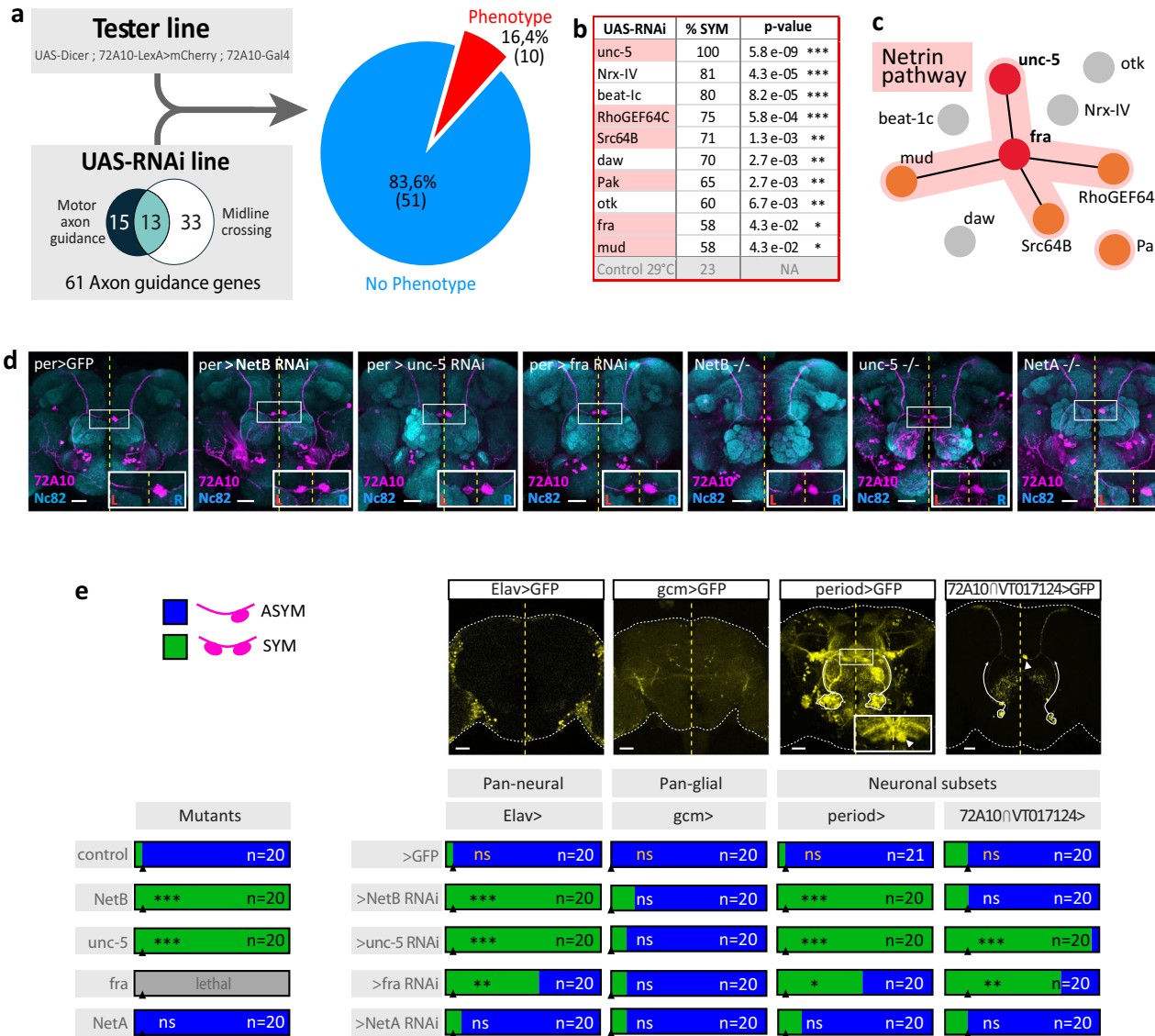

**Fig. 3 | The NetB pathway controls asymmetric development of the H-neurons.**
**a** RNAi screen strategy and validated candidates. The tester line allows co-expression in H-neurons of both RNAi and a fluorescent reporter in the asymmetric circuit. **b** Among 61 candidate genes known to mediate axon guidance and/or midline crossing, 10 showed a statistically significant increase in the proportion of SYM flies when their function is blocked ($n \geq 20$ brains/condition. *P*-values are calculated with Pearson's Chi-squared test and Benjamini and Yekutieli multiple comparison correction. Significance threshold are: *, <0.05; **, <0.01; ***, <0.001). **c** GO term analysis with String database (v11) indicates that GO terms GO:0005042 "netrin receptor activity" and GO:0038007 "netrin-activated signalling pathway" are the most enriched terms for Molecular function and Biological process respectively (strength = 3.14 and 2.97). Red dots on the diagram indicate Netrin receptors, orange dots indicate Netrin pathway effectors and black lines indicate evidence for protein-protein interaction in *Drosophila*. **d** Confocal images of representative adult brains displaying most frequently observed phenotype (see **e**) following *netB*, *unc-5* and *fra* RNAi mediated loss-of-function in *period-Gal4* neurons or in *netB*, *netA* and *unc-5* mutants. Brain neuropils are labelled with the Nc82 antibody (cyan) and H-neurons are labelled with the *72A10-lexA* driver (magenta).

Scale bars, 30 μm. Insets show enlargements of the AB region and the yellow dotted line represent the brain midline. **e** Domain requirement and phenotype of Netrin pathway genes. A collection of Gal4 lines were used to drive expression of RNAi lines targeting Netrin pathway genes. Confocal brain images illustrate the expression pattern of each Gal4 line. Scale bars, 30 μm. *unc-5* and *fra* but not *netB* are required in H-neurons. Bars represent frequencies of ASYM (blue) and SYM (green) phenotypes in adult flies. Black triangles represent control SYM frequencies for each set of experiment. n represent the number of brains analysed. *P*-values are calculated with Pearson's Chi-squared test and Benjamini and Yekutieli multiple comparison correction. *P*-values are: wildtype vs NetBΔ: 3,31E-08; wildtype vs unc-5 MI05371: 3,31E-08; wildtype vs Elav>NetB RNAi: 3,01E-08; wildtype vs Elav>unc-5 RNAi: 3,01E-08; wildtype vs Elav>fra RNAi: 1,48E-03; per>GFP vs per>NetB RNAi: 3,01E-08; per>GFP vs per>unc-5 RNAi: 3,01E-08; per>GFP vs per>fra RNAi: 3,94E-03; 72A10 ∩ VT017124 > GFP vs 72A10 ∩ VT017124 > unc-5 RNAi: 3,70E-04; 72A10 ∩ VT017124 > GFP vs 72A10 ∩ VT017124 > fra RNAi: 1,36E-04. Significance threshold are: *<0.05; **<0.01; ***<0.001. *P*-values in yellow correspond to comparisons between Gal4 controls and mutant control. Source data are provided as a Source Data file.

H-neurons (Fig. 6a). Lineage study[53] indicates that the LALv1A lineage is composed of 29 neuron types, including the H-neuron type. We conclude that NetB is active in one or more of the 28 non-H-neuron type(s), specifically on the right side (Fig. 6a). In both vertebrates and *Drosophila*, the conserved NetB pathway is required for guiding axons through repulsion/attraction mediated by its receptors Unc-5

and Fra. We propose that unilateral activity of NetB in the right LALv1A lineage acts non-autonomously to promote Right-specific attraction of H-neurons towards the right AB (Fig. 6a). Additionally, maintaining NetB signalling specifically on the right side could contribute to the maintenance/stabilization of the right hand of the initially bilateral circuit.

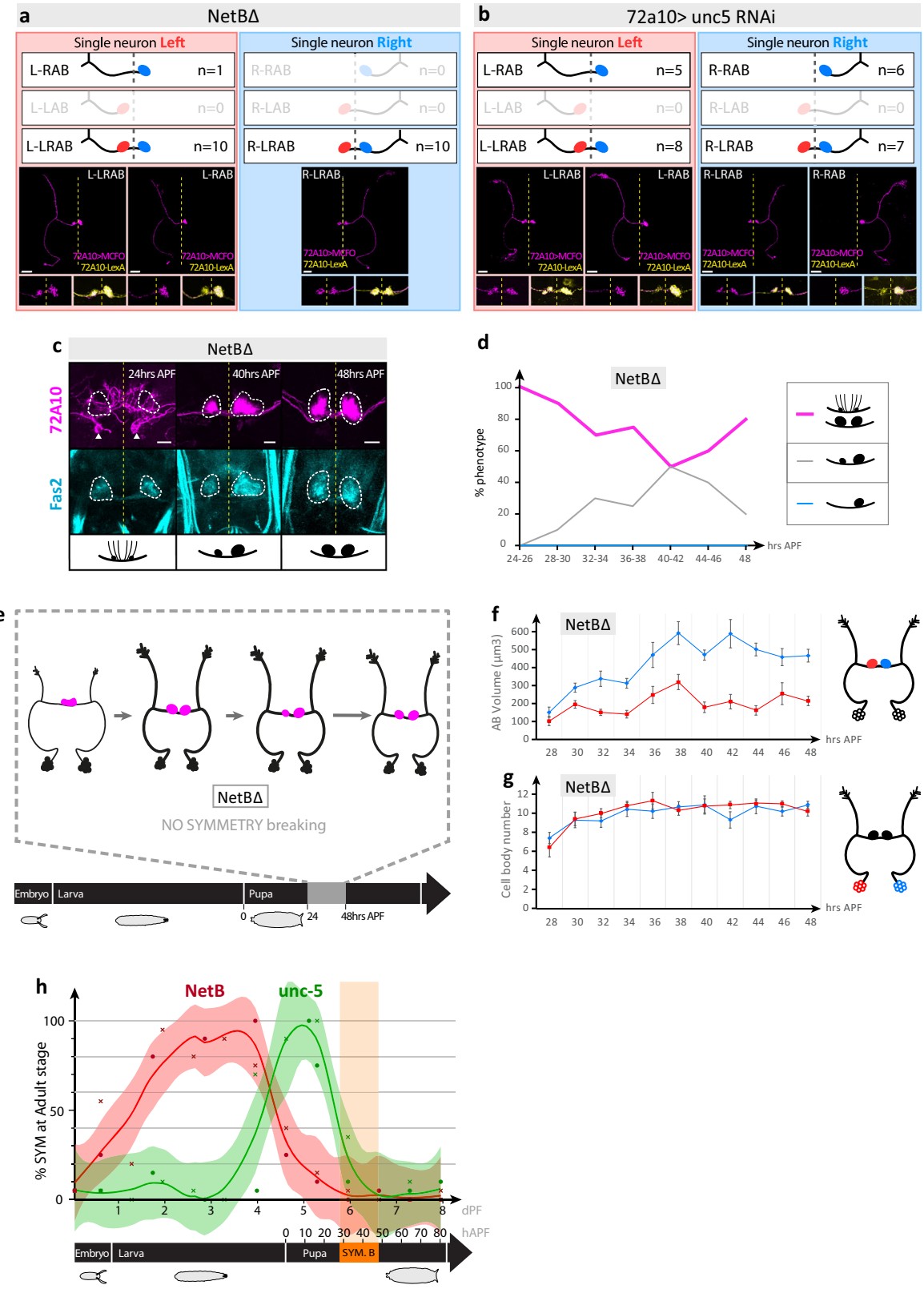

To gain insights into the functional importance of NetB-induced laterality, we sought to investigate whether mutant flies with symmetric brains recapitulated the olfactory long-term memory (LTM) impairments observed in the naturally occurring SYM ones[16]. We thus tested memory performance of *per-GAL4 > unc-5 RNAi* flies using the same aversive associative paradigm (odour/electric shock pairing)[16]. When raised at 18°, which favours robust LTM scores in control flies,

the *per-GAL4 > unc-5 RNAi* condition still yielded 100% of flies with symmetric brains (Fig. S5). When tested for their LTM formation ability, i.e., measuring memory performance 24 h after a spaced training, *per-GAL4 > unc-5 RNAi* flies showed a strongly decreased memory score as compared to their genotypic controls (Fig. 6b, left panel). However, the memory measured 24 h after a massed training, which is a distinct consolidated memory from LTM, was not different

**Fig. 4 | Single neuron morphology and development of H-neurons in NetB pathway loss-of-function. a, b** Single-cell clone analysis of H-neurons in *netB*[4] **a** and *72A10-Gal4 > unc-5-RNAi* **b** flies. Confocal images of representative single neuron types are labelled as in Fig. 1j–m and their occurrence indicated as n numbers. Insets at the bottom show higher magnification of AB region with representative single neuron types (magenta) and the whole 72A10 neuron circuit (yellow). Dotted line represents the brain midline. Scale bars, 30 μm. **c** Development of the *netB*[4] mutant H-circuit during pupal stages. In contrast to wild-type ASYM flies, *netB*[4] H-neuron projections remain in their initial symmetrical state (compare with Fig. 2a). Confocal images of representative *netB*[4] mutant H-neurons (magenta) stained with anti-Fas2 antibody (cyan) during pupal stage. ABs (white dotted lines) and noduli (white arrows) are shown. Phenotype categories are defined as in Fig. 2a. Scale bars, 10 μm. **d** Evolution of the *netB*[4] mutant H-circuit during pupal development (28–48 h APF) showing that in contrast to wild-type, the decrease of the symmetric phenotype does not reach a plateau as in wild-type flies (compare with Fig. 2b) but instead increases in a second phase, so no occurrence of the "right" phenotype is observed in *netB*[4] mutant flies. (*n* = 21 brains for 28–30 h APF and *n* = 20 brains for 32–34, 36–38, 40–42, and 44–46 h APF representing the merging of 2 adjacent data points shown in **f**; *n* = 10 for 48 h APF. Source data are provided as a Source Data file. **e.** Summary of symmetrical H-neuron pattern formation in *netB*[4] mutant flies during pupal stage. **f, g** Morphometric analysis of NetB mutant's H-neurons during pupal development (28–48 h APF), showing the evolution of the volume of left and right AB projections **f**, and the number of left and right H-neuron cell bodies **g**. AB projections volume: *n* = 11 brains for 28 h APF; n = 10 for all other time points. H-neuron cell bodies (L/R): *n* = 10/10 for 28, 30, 46, and 48 h APF; *n* = 9/10 for 32 and 38 h APF; *n* = 10/9 for 34 and 42 h APF; *n* = 9/9 for 36, 40, and 44 h APF. Data are presented as mean ± standard error of the mean (SEM). **h** Temporal requirement for *netB* and *unc-5* function during development, showing sequential gene activities taking place before symmetry breaking. Data represent the frequency of adult SYM phenotype following temporally restricted *netB* and *unc-5* loss-of-function. Solid red and green line curves represent local regression of the frequency data obtained with 2 different methods: either ON > OFF/OFF > ON RNAi expression (data plotted as red and green circles) or using 24 h pulse RNAi expression (data plotted as red and green crosses). 20 brains are analysed for each data point and for both methods (see text, source data file and Methods for details). Shaded areas around solid black line curves represent 95% confidence interval. dPF, days post-fertilisation (25 °C); hrs APF, hours after puparium formation (25 °C). Source data are provided as a Source Data file.

---

from the genotypic controls (Fig. 6b, middle panel). We also tested memory at earlier time points. 3 h after a single cycle of training, *per-GAL4 > unc-5 RNAi* flies showed a similar memory score as their genotypic controls, either with or without cold anaesthesia, showing that both the labile and anaesthesia-resistant components of the memory[54–56] were preserved in symmetric flies (Fig. 6b, right panel). To further support a role of H-neurons in LTM, we performed memory tests using courtship conditioning, an alternative associative paradigm in which males learn to suppress their courtship behaviour upon exposure to unreceptive mated females[57]. Symmetric *per-GAL4 > unc-5 RNAi* males exhibited significantly reduced LTM memory indices (Fig. 6c, left panel) as evidenced by a poor suppression of courtship behaviour 24 h after spaced training. In contrast, these flies displayed normal short-term memory (STM) when tested 30 min after a single training procedure (Fig. 6c, right panel). Altogether, these experiments show that *unc-5-RNAi* symmetric flies are specifically defective in LTM formation.

## Discussion

In this study, we identify the first genes controlling brain laterality in *Drosophila*. We also reveal a previously unknown unilateral activity of the NetB pathway, required for building an asymmetrical neuronal circuit involved in long-term memory. These results show a simple mechanism for handed circuit assembly, involving an extra layer of patterning of an axon guidance pathway through lateralization of its ligand's activity. Whether this could represent a general principle is suggested by expression data from the cichlid fish *Perissodus microlepis* showing asymmetric expression of *net1n* in the hindbrain[58]. Furthermore, some human *netG* isoforms show asymmetrical expression in the anterior cingulate cortex[59], and an enhanced expression of *netG* is observed in bipolar patients[59], a condition which can be associated with a decrease in brain laterality. The asymmetry of ABs and H-neurons share some common features with the vertebrate habenula[60,61], which also processes aversive responses from a variety of stimuli. The habenula shows anatomical and functional LR asymmetry, in particular in zebrafish, where the processing of olfactory stimuli takes place specifically in the right habenula, receiving input from the olfactory bulb through specialized neurons, the mitral cells[62,63]. Future work comparing different paradigms, including *Drosophila* H-neurons, will help determine the evolutionary conservation and common principles guiding the assembly and function of lateralized neuronal circuits.

The fact that the *myo1D* chiral factor does not play any role in *Drosophila* brain laterality, together with current data in vertebrate models and human[1], support the general idea that body and brain asymmetry are not coupled in animals. For example, people with situs inversus of their visceral organs show a normal ratio of left-handers[64]. Hence, brain laterality mostly relies on unique symmetry-breaking mechanisms, likely a consequence of their specific cell types (neuron, glia), and organization (non-epithelial tissue) compared to visceral organs. Another specific feature of brain asymmetry is that it is less robust than body asymmetry. For example, the habenula is randomized in 5–10% of zebrafish, left-handedness in human is present in 10% of the population, and symmetry of the *Drosophila* AB is observed in 5% of the flies. Understanding the basis and function of these minority-generating fluctuations represent an important goal to fully understand brain asymmetry and how it could contribute to e.g., individuality and fitness. We anticipate that the H-neuron model will help identify key signalling pathways and mechanisms underlying the logic and specificity of brain asymmetry as well as individual and population level behaviours relying on a lateralized nervous system.

## Methods

### *Drosophila* strains and genetics

For the RNAi screen, lines targeting genes involved in midline[39] or motor[38] axon guidance were selected and ordered from the Vienna *Drosophila* RNAi Centre (VDRC, Vienna, Austria) and the Bloomington *Drosophila* Stock Center (BDSC, Bloomington, IN, USA). The list of RNAi lines used in the screen is provided as a Supplementary Data (Data S1). Other *Drosophila* lines are the following. Stocks obtained from the Bloomington *Drosophila* Stock Center: 72A10-Gal4 (#48306); 72A10-LexA (#54191); 72A10-p65AD (#70799); Elav-Gal4 (#8765); gcm-Gal4 (#35541); NetB-Gal4 (#76730); per-Gal4 (#7127); VT017124-GDBD (#75461); 38D01-Gal4 (#49996); NetAΔ, NetB-TM (#66880); NetB::GFP BA00253 (#50794); NetB::GFP MI10467 (#67644); unc-5::GFP MI05371 (#60547); NetAΔ, (#66878); NetBΔ (#66879); unc-5 MI05371 (#42316); fra[3] (#8813); fra[4] (#8743); 13xLexAop2-6xEGFP (#52265 and #52266); 13xLexAop2-6xmCherryHA (#52271 and #52272); 20XUAS-6xEGFP (#52261 and #52262); 20XUAS-6xmCherryHA (#52267 and #52268); UAS-Stinger (#84278 and #65402); UAS-beat-Ic RNAi (#64528); UAS-daw RNAi (#50911); UAS-fra RNAi (#31664); UAS-fra RNAi (40826); UAS-mud RNAi (#35044); UAS-NetA RNAi (#31288); UAS-NetB RNAi (#25861); UAS-NrxIV RNAi (#38192); UAS-otk RNAi (#67966); UAS-Pak RNAi (#62201); UAS-RhoGEF64C RNAi (#77431); UAS-Src64B RNAi (#51772); UAS-unc-5 RNAi (#33756); w + [CS][54]; w1118 [CS][54]; FRT19A (#1709); hsFlp, FRT19A, tubP-Gal80ts (#5133); MCFO1 (#64085); MCFO2 (#64086); MCFO4 (#64088); MCFO5 (#64089); Df(3 L)99 (#1576)[65]; tubP-Gal80ts (#7019); UAS-Dicer2 (#24646 and #24651); UAS-P35 (#5073)[66]. Stock obtained from the Vienna

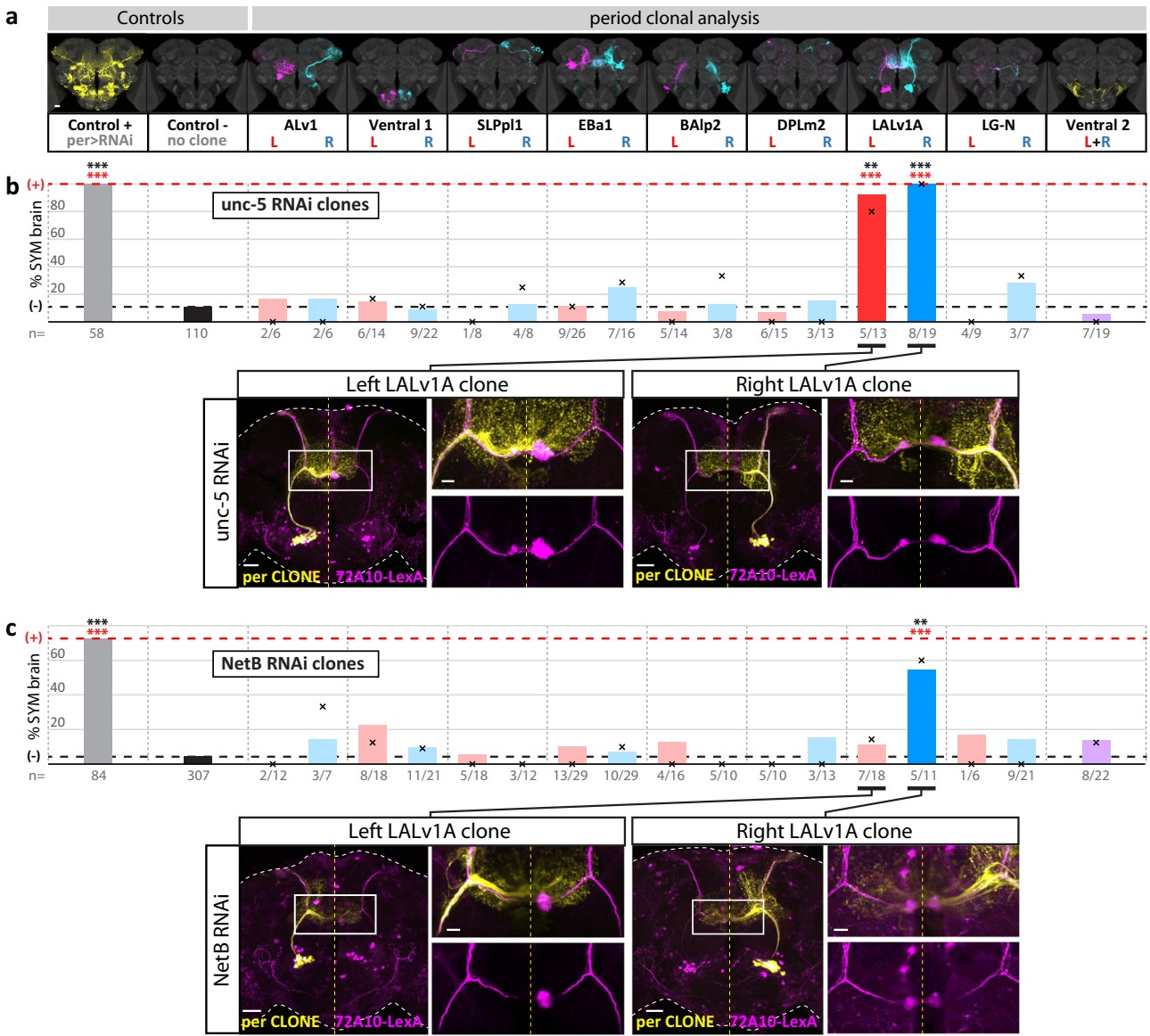

**Fig. 5 | NetB is required asymmetrically in the right LALv1A lineage.**
**a** Generation of unilateral clones expressing RNAi against *netB* or *unc-5*. Representative left and right clones generated in 9 stereotyped clonal groups of neurons derived from the *period-Gal4* driver, using the MARCM method, including 4 known lineages (ALv1, EBa1, LALv1A, BAlp2), 3 lineage subsets (SLPpl1, DPLm2 and the LG-N neurons) and 2 other neuronal groups (hereafter referred to as Ventral1 and Ventral2). Clones are identified by a fluorescent marker. Scale bar, 30 μm. **b** *unc-5* RNAi mediated loss-of-function in left or right LALv1A lineage induces a loss of asymmetry phenotype. Representative confocal images of phenotypes observed in brains expressing *unc-5* RNAi in left (left panel) or right (right panel) LALv1A neurons (scale bars, 30 μm), with enlarged views of the boxed AB region (scale bars, 10 μm). Source data are provided as a Source Data file. **c** *netB* RNAi mediated loss-of-function in right LALv1A lineage induces a loss of asymmetry phenotype. Representative confocal image of phenotypes observed in brains expressing *netB* RNAi in left or right LALv1 neurons (scale bars, 30 μm), with enlarged views of the boxed AB

region (scale bars, 10 μm). In **b**, **c**, the black crosses represent the observed frequencies of SYM phenotypes in brains with a single clone. Coloured bars represent the observed frequencies of SYM phenotypes in brains with single and multiple clones. Dotted black and red lines represent the percentage of SYM phenotype in negative and positive controls, respectively. Two-tailed Fisher-test pValues with Benjamini–Hochberg correction are: *unc-5* RNAi positive control: 2,83E-32; *unc-5* RNAi Left LALv1A clone (single/single+multiple): 7.21E-03/1.32E-08; *unc-5* RNAi Right LALv1A clone (single/single+multiple): 1.55E-06/4.93E-14; *netB* RNAi positive control: 9.95E-38; *netB* RNAi Right LALv1A clone (single/single+multiple): 9.43E-03/6.86E-05. Significance threshold for *p*-values are: <0.05; ** <0.01; *** <0.001. Black stars, brains containing single clones; red stars, brains containing single and multiple clones). Numbers below bars represent: the number of brains analysed (controls), and the number of "single-clone containing"/"single + multiple-clone containing" brains analysed (clones). Source data are provided as a Source Data file. All calculated *p*-values are listed in Data S3.

*Drosophila* Resource Center: UAS-NetA RNAi (#330207). Stock obtained from the Kyoto *Drosophila* Stock Center: NetB::GFP CPTI168 (#115011). Stock generated in our laboratory: Myo1DK2[24]. Stock kindly provided by Darren Williams: NetAΔ, NetBmyc[51]. The list of fly's genotypes used in each figure is provided as a Supplementary Data (Data S2). UAS-unc-5 RNAi (#33756) and per-Gal4 lines used for memory test were isogenized with Canton-S background by outcrossing them for six generations to a w1118 [CS] (Canton-Special) stock.

## Morphometric analysis on adult and pupal stages
Analysis was done on adult flies or pupal stages both raised at 25 °C. Brains of adult female and males were dissected in cold PBS 2 to 5 days post emergence. For pupal stages, flies were allowed to lay eggs on agarose plates for 24 h then plates were transferred in tubes containing fly media. White pupas were sorted from tubes and dissected 24-to-48 h APF in cold PBS. Pupal stage brains were immunostained with anti-GFP antibody.

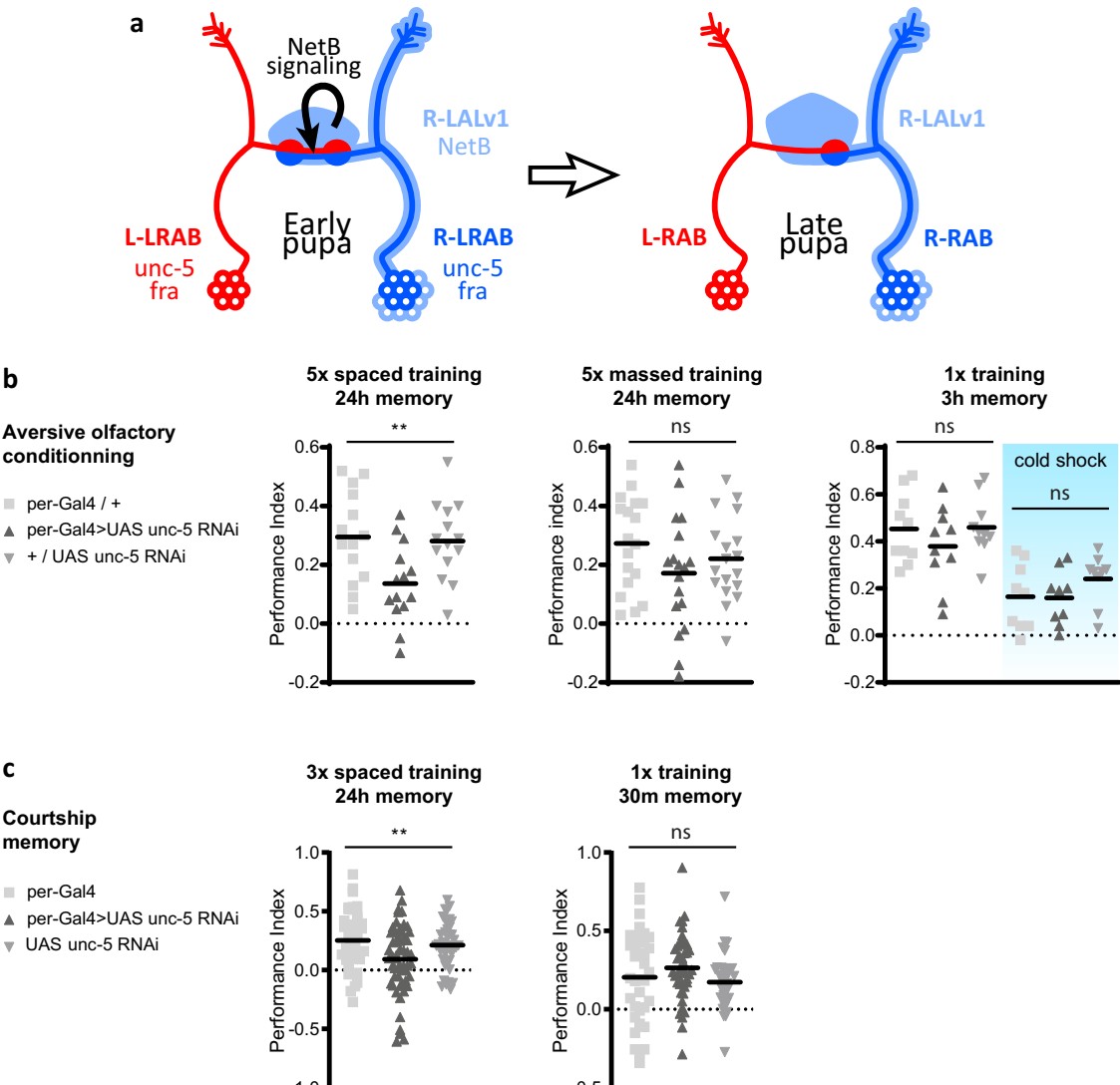

**Fig. 6 | Model of NetB pathway function in H-neuron laterality and memory performance. a** Model of NetB pathway activity in H-neuron laterality. At early pupal stage, NetB expressed in the right LALv1A lineage (light blue) signals to left and right H-neurons (dark blue) expressing Fra and Unc-5 receptors. NetB instructs neurons through right-specific attraction and/or synapse maintenance. **b** Memory performance after olfactory aversive conditioning of flies with symmetric brains (*per-GAL4 > UAS-unc-5 RNAi*) as compared to genotypic control groups (*per-GAL4/+* and *+/UAS-unc-5 RNAi*). Symmetric flies are defective for long-term memory performance (5× spaced training; *n* = 14 per group), but not for another form of consolidated memory (5× massed training; *n* = 18 per group), nor for labile or anaesthesia-resistant memory (1× training; no cold shock: *n* = 10 per group; cold shock: *n* = 9 per group). Horizontal bars represent mean value. Groups were compared using 1-way ANOVA followed by Tukey's pairwise comparisons. *P*-values are:

5× spaced training: 0.0081; 5× massed training: 0.2055; 1× training: 0.4123; 1× training + cold shock: 0.3165. Source data are provided as a Source Data file. **c** Memory performance Indices after courtship conditioning of flies with symmetric brains (*per-GAL4 > UAS-unc-5 RNAi*) as compared to genotypic control groups (*per-GAL4* and *UAS-unc-5 RNAi*). Symmetric flies are defective for long-term memory performance (LTM 24 h; left), but not for short-term memory (STM 30 min; right). Number of trained males tested for each genotype: *per-GAL4: n* = 40 (LTM) and 35 (STM); *UAS-unc-5 RNAi: n* = 44 (LTM) and 33 (STM); *per-GAL4 > UAS-unc-5 RNAi*: 56 (LTM) and 43 (STM). Horizontal bars represent mean value. Groups were compared using 1-way ANOVA followed by Tukey's pairwise comparisons. P-Values are: 3× spaced training: 0.0054; 1× training: 0.2151. Source data are provided as a Source Data file. Significance thresholds for ANOVA p-values are: * <0.05; ** <0.01; *** <0.001. n.s. stands for not significant. All calculated *p*-values are listed in Data S3.

## Single cell labelling

MultiColor FlpOut (MCFO)[23] lines were crossed at 25 °C and their 2 to 5 day-old adult female and male progeny with correct genotype heat-shocked in a water bath at 37 °C during 15 to 45 min. For wild-type condition MCFO1, MCFO2, MCFO4 and MCFO5 were used. For NetB mutant and unc-5 RNAi conditions, lines carrying hs-FLPG5.PEST transgene from MCFO1 first chromosome and Flip-out cassettes from MCFO4 third chromosome were built. Genotype for these lines is "hs-FLPG5.PEST, NetBΔ; <MCFO4 Flip-out cassettes >" and "hs-FLPG5.PEST;; UAS-unc-5 RNAi (#33756), <MCFO4 Flip-out cassettes >" respectively. For all conditions, virgins from MCFO and MCFO-derived lines were crossed to males with genotype "72A10-LexA, 13xLexAop2-

6xmCherryHA/CyO; 72A10-Gal4/TM3, sb". Brains were dissected in cold PBS 2 to 3days after heat-shock and immunostained with Flag and V5 antibodies.

## RNAi screen and mutant/Gal4 screen

For RNAi screen, virgin female flies from "Tester line" were crossed with males from a collection of UAS-RNAi lines and raised at 29 °C. Tester line genotype is: UAS-Dicer2; 72A10-LexA, 13xLexAop2-6xmCherryHA/CyO; 72A10-Gal4/TM6b. For Gal4 screen: virgin female "Tester lines" with genotype "UAS-Dicer2; 72A10-LexA, xxx-Gal4/CyO; 13xLexAop2-6xmCherryHA" (with xxx-Gal4 = Elav-Gal4, gcm-Gal4 or per-Gal4) were crossed with NetB, fra, unc-5 or NetA RNAi or "Tester

lines" with genotype "UAS-Dicer2; 72A10-LexA, 13xLexAop2-6xmCherryHA; UAS-xxx RNAi/TM6b" (with xxx RNAi = NetB, fra, unc-5 or NetA RNAi) were crossed with males from "72A10-p65AD/CyO; VT017124-GDBD/TM3" split-gal4 line. For mutant and Gal4 screen flies were raised at 25 °C. For all experiments at least 20 F1 adult flies, half males and half females were dissected 2 to 5 days post emergence. Brains were mounted in PBS on microscope slides and position of their asymmetrical body scored by inspection of H-neuron projections on an epifluorescence microscope (Leica DMR).

## Immuno-labelling

After dissections, brains were fixed 20 min at room temperature in 4% PFA-PBS then washed 3 times with PBS 0.5% Triton before performing immunostainings. For adult single cell labelling and morphometric analysis on pupal stages brains were fixed over-night at 4 °C in 1%PFA-PBS then washed 3 times with PBS Triton 0.5% Triton before performing immunostainings. For immunostainings, brains were blocked 1 h at 4 °C in PBS Triton 0.5% - BSA 0.1% then incubated with primary antibody diluted in PBS Triton 0.5% overnight at 4 °C. Brains were washed 3 times with PBS Triton 0.5% Triton followed by incubation with secondary antibody diluted in PBS Triton 0.5% overnight at 4 °C and finally washed 3 times with PBS Triton 0.5%. Primary and secondary antibody Incubation time was expanded to 2 days for adult single cell labelling and morphometric analysis on pupal stage. Brains were mounted on microscope slides in Vectashield (Vectorlabs) mounting media for observation at the confocal. Primary antibody references and dilutions used are: Mouse anti-Fas2 (Developmental Studies Hybridoma Bank - Ref 1D4 – 1:100), Mouse anti-brp (Nc82) (Developmental Studies Hybridoma Bank - Ref nc82 – 1:100), Goat anti-GFP (Antibodies-online – Ref ABIN100085 – 1:200), Mouse anti-V5 (Invitrogen – Ref 46-0705 (new Ref R960-25) – 1:300), Rat anti-Flag (Novus – Ref NBP1-06712SS – 1:300), Rabbit anti-Flag (Sigma – Ref F7425 – 1:300). Secondary antibody references and dilutions used are: Donkey anti-mouse Alexa Fluor 647 (Invitrogen - Ref A31571 – 1:500), Donkey anti-mouse Alexa Fluor 488 (Invitrogen - Ref A21202 – 1:500), Donkey anti-rat Alexa Fluor 647 (Invitrogen – Ref A21247 – 1:500), Donkey anti-rabbit Alexa Fluor 647 (Invitrogen - Ref A31573 – 1:500), Donkey anti-goat Alexa Fluor 488 (Invitrogen – Ref A-11055 – 1:500).

## RNAi sensitive period analysis

For NetB and unc-5 RNAi sensitive period, virgin females of genotype "UAS-Dicer2; tubP-Gal80ts; UAS RNAi" were crossed with males of genotypes "UAS-Dicer2; 72A10-LexA, Elav-Gal4/CyO; 13xLexAop2 6xmCherryHA". Two different methods were used. For ON > OFF/OFF > ON method, flies are allowed to lay eggs on agarose plates for 2 h in an incubator set at either 22 °C ("OFF-ON" curves) or 29 °C ("ON-OFF" curves), then plates were transferred in a tube containing fly media in an incubator set at the same temperature. For each "day D" point of the curves, temperature changes from 22 °C to 29 °C (OFF > ON) or 29 °C to 22 °C (ON > OFF) were made Dx24 + 1 h after the onset of the egg-laying period. For "24-h pulse" experiments, tubes were kept at 22 °C then moved to 29 °C incubator Dx24 + 1 hour after the onset of the egg-laying and back to 22 °C incubator Dx24 + 24 + 1 hour after the onset of the egg-laying. For both methods, female and male adults with appropriate genotype were dissected in cold PBS 2 to 3 days after their emergence. Curves representing sensitive period along development were derived from local regression analysis of the data from both methods using R v4.1.1 and ggplot2 v3.3.2 package.

## period-Gal4 clonal analysis

For unc-5 RNAi clones, virgin females of genotype "FRT19A; 72A10-LexA, 13xLexAop2 6xmCherryHA/CyO; UAS-unc-5 RNAi" were crossed with males of genotypes "hsFlp, tubP-Gal80, FRT19A; per-Gal4, 20XUAS-6xEGFP/CyO". For NetB RNAi clones, virgin females of genotype "FRT19A; 72A10-LexA, 13xLexAop2 6xmCherryHA/CyO; UAS-NetB

RNAi" were crossed with males of genotypes "hsFlp, tubP-Gal80, FRT19A; per-Gal4, 20XUAS-6xEGFP/CyO; UAS-Dicer2". For both conditions, flies were allowed to lay eggs on agarose plates for 16-to-24 h then plates were transferred in a tube containing fly media. 1-to-2 h following the transfer, tubes were heat-shocked in a 37 °C water bath for 2-to-3 h. Tubes were finally transferred in a 25 °C incubator and brains of adults with appropriate genotype were dissected in cold PBS 2 to 3 days after their emergence. Per-Gal4 has already been reported as a specific marker for ALv1/BAla1, EBa1/DALv2, LALv1/BAmv1 lineage marker[67–72]. We found that early recombination events using the per-Gal4 driver labels 9 stereotyped clonal groups of neurons. Based on their morphology we indeed identified among them the 3 lineages already identified but report here that only LALv1A (Notch ON) hemi-lineage is labelled by per-Gal4[21,73,74]. A group formed of 2 cells (one in each hemisphere) corresponding to LG-N (LAL-GA-NO1) neurons[17] were also labelled, independently from the LALv1A lineage, from which they are sought to belong[21]. We also identified labelling in what appears as the full BAlp2 lineage[67,70] and subsets of the SLPpl1/DPLl1[67,73,74] and DPLm2[67,70] lineages. Finally, we found labelling of two groups of ventrally located neurons (Ventral1 and Ventral2) whose morphology does not match any previously reported neuronal lineage. Ventral2 group of neurons is the only one for which distinction between left and right neurons was impossible as their cell body are centered along the midline and their projections appear symmetrical. In the F1 progeny, identity and number of clones (green channel) as well as 72A10 neuron asymmetry (red channel) were scored in female brains. Females showing no labelled clones in green channel were used as negative controls while male brains that do not carry "tubP-Gal80" transgene were used as positive controls. Scoring is done first using an epifluorescence microscope (Leica DMR) then Zeiss 780 confocal microscope for brains for which scoring appeared ambiguous with epifluorescence microscope. Brains observed at the confocal microscope are immunostained with Nc82 antibody. Statistical analysis was done in two steps. First, on scored brains containing single clones (single clone brains). This first analysis indicated that 2 clone types induced a significant SYM phenotype when expressing *unc-5 RNA* (Left and Right LALv1A), while only 1 clone type induced a significant phenotype when expressing *netB RNAi* (Right LALv1A) (Data S3). In a second step, brains containing multiple clones were considered. The scoring of brains showing SYM phenotype and containing multiple clones was corrected according to the identity of clones showing statistical support in the statistical analysis done in the first step (corrected multiple clone brains). Accordingly, SYM brains containing multiple unc-5 RNAi clones among which Left or Right LALv1A clones, were scored as brains containing a unique Left or Right LALv1A clone, respectively, and SYM brains containing multiple NetB RNAi clones among which Right LALv1A clone, were only scored as brains containing a unique Right LALv1A clone. All other brains containing multiple clones with none of them showing statistical support in the first step, were otherwise scored for each individual clone they contained (non-corrected multiple clone brains). Finally, a second statistical analysis was conducted on all analysed brains using scoring for brains containing single clones and scoring for brains containing multiple clones (single clone brains + corrected multiple clone brains + non-corrected multiple clone brains).

## Memory tests

**Aversive memory assay.** For olfactory associative memory assays and the corresponding controls, flies were raised at 18 °C with a 12 h/12 h light/dark cycle. All behaviour experiments were performed on young adult female and male flies (1–3 day-old), in dedicated rooms where temperature was kept at 25 °C and humidity at 80%.

Aversive associative conditioning was performed in custom-designed barrel-type apparatus as previously described[16], which allows the parallel conditioning of six groups of flies in six independent cylindrical chambers.

The odorants 3-octanol and 4-methylcyclohexanol, diluted in paraffin oil at a final concentration of 0.29 g L-1, were used for conditioning and for the test of memory retrieval.

Groups of 30–50 flies were subjected to either single-cycle, five spaced cycles or five massed cycles of aversive associative olfactory conditioning[55]. One cycle of aversive olfactory conditioning consisted in the following sequence: throughout the conditioning protocol, flies were submitted to a constant air flow at 0.6 L min$^{-1}$. After 90 s of habituation, flies were first exposed to an odorant (the CS + ) for 1 min while 1.5 s-long pulses of 60-V voltage was applied every 5 seconds on the copper grid covering the walls of the barrel chambers (12 pulses in total); flies were then exposed 45 s later to a second odorant without shocks (the CS-) for 1 min. 3-octanol and 4-methylcyclohexanol were alternately used as CS + and CS-. Massed training consisted in five consecutive cycles. Spaced training consisted in five cycles spaced by rest intervals of 15 min during which flies were kept in the conditioning barrel.

The memory test was performed in a T-maze apparatus. Each of the two arms of the T-maze was connected to a bottle containing one odorant (either 3-octanol or 4-methylcyclohexanol) diluted in paraffin oil at the same concentration as for conditioning. The global air flow from both arms of the T-maze was set to 0.8 L min$^{-1}$. Flies were given 1 min in complete darkness to freely move within the T-maze. Then flies from each arm were collected and counted. The repartition of flies was used to calculate a memory score as (NCS + -NCS-)/(NCS + + NCS-). A single performance index value is the average of two scores obtained from two groups of genotypically identical flies conditioned in two reciprocal experiments, using either odorant as the CS + . Therefore, each value of performance index involves 60–100 flies. Performance index ranges between −1 and +1, the value of 0 (equal repartition) corresponding to the absence of memory. The indicated number of repeats 'n' is the number of independent performance index values for each genotype.

Memory performance was assessed 24 h (±2 h) after spaced or massed conditioning, and 3 h ± 30 min) after single-cycle conditioning. To test for ARM performance after 1× training, flies were subjected to cold treatment exposure (4 °C for 2 min) 1 h before testing.

**Courtship memory assay.** For courtship suppression assays, flies were raised at 25 °C with a 12 h/12 h light/dark cycle. All experiments were performed with flies cantonized for 6 generations, in a dedicated room where temperature was kept at 23–25 °C and humidity at 60–80%.

Briefly, virgin male flies were collected between 0 and 4 h after eclosion and transferred to individual glass food vials, where they were aged for 5 days before behavioural training with pre-mated females. Canton S virgin females were collected in parallel and kept in normal food vials in groups of 10. 16 h before the beginning of training, females were pre-mated with >5 day-old Canton S males previously housed in groups of 15.

For short-term memory (STM) training, individual males were placed in individual small (16 × 100 mm) glass food tubes either with (trained males), or without (naive males), a mated female for 1 h. Female were then removed from the tester tubes and males kept in isolation for 30 minutes, followed by STM testing in 2.5 cm diameter courtship chambers.

For long-term memory (LTM) training, spaced training was performed in the same food tubes as for STM, but males were exposed to three different mated females, for 2 h each, with a resting interval of 30 min. LTM was assayed 24 h after training.

Courtship behaviours were recorded for 12.5 min, and courtship indices (percentage of time spent by males on courting) were automatically extracted from t = 2.5 min onwards, using a custom-built Fiji algorithm[75]. Performance Indices (or courtship suppression indices) were calculated for each tested male as follow: 1–(CI Trained / CI Naive), where CI Trained represents the courtship index of the trained fly, and CI Naive represents the mean courtship index of the naive flies, respectively.

## Confocal acquisition and image analysis
Confocal imaging was performed using Zeiss 780 or 880 confocal microscopes. Fiji v1.52p was used for visualisation and CMTK plugin used to align period-Gal4 clone labellings on the Tefor reference brain. 3D reconstruction was performed with IMARIS software (9.6.1) to measure the AB volume and length of the different part of H-neuron projections.

## Identification of H-neurons in a connectome dataset
Connectome dataset from a single brain from an adult female individual[76] was searched for neurons matching H-neurons morphology and projecting in the right asymmetrical body using the neuprint v1.2.1 dataset and "Find neurons" tool (neuprint.janelia.org). Neurons were then visualised using the neuprint "vizualisation/skeleton" tool.

## Statistics and reproducibility
Data was collected using Microsoft Excel 2016 and statistical analysis was conducted with R 4.1.1 or GraphPad Prism. Statistical tests used are: AB volume and cell body number analysis Left vs Right: Two-tailed Wilcoxon matched-pairs signed rank test (Prism v9.3.1); AB volume analysis Left+Right total and cell body number analysis Left vs other Left/Right vs other Right: Krustal/Wallis test with Dunn's multiple comparisons correction (Prism v9.3.1); AB volume analysis Left/Right ratios: Two-tailed Mann-Whitney test (Prism v9.3.1); RNAi screen/mutant, alleles and RNAi phenotypic analysis: Pearson's Chi-squared test with Benjamini & Yekutieli multiple comparisons correction (R); period-Gal4 clonal and single cell analysis: Two-tailed Fisher's Exact Test with Benjamini & Yekutieli multiple comparisons correction (R); Memory test: Performances from different groups (mutant and genotypic controls) were statistically compared using 1-way ANOVA followed by Tukey pairwise post-hoc comparisons (Prism v9.4.1). ANOVA results are presented as the value of the Fisher distribution $F(x,y)$ obtained from the data, where x is the number of degrees of freedom between groups and y is the total number of degrees of freedom for the distribution. P-Values for each test are provided as a Supplementary Data (Data S3).

All confocal pictures presented in the figures show a representative specimen from at least 3, and typically 5, acquisitions.

## Reporting summary
Further information on research design is available in the Nature Portfolio Reporting Summary linked to this article.

## Data availability
Source data are provided with this paper.

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

## Acknowledgements

We wish to thank Thomas Kidd, Iris Salecker, Darren Williams, and the Vienna, Bloomington and Kyoto stock centers for materials and fly stocks. We thank Laura Guédon for technical help, the iBV bioinformatics and imaging platforms for help with statistical analysis and image acquisition, Bruno Hudry for suggestions and materials, Nathan Lerousseau and Wendy Marcantonio for help in early explorative efforts, Jean-Marc Gambaudo and Sylvie Mellet for their support, and members of the SN laboratory for discussions. Work in SN laboratory is supported by Agence Nationale pour la Recherche (ANR-17-CE13-0024 (SN); ANR-20-CE13-0004 (SN)), Fondation pour la Recherche Médicale (FRM; EQU201903007825 (SN)), Université Côte d'Azur (UCA), Centre National pour la Recherche Scientifique (CNRS), Institut National pour la Recherche Médicale (Inserm), LABEX SIGNALIFE (ANR-11-LABX-0028-01 (SN)).

## Author contributions

Conceptualisation: F.L. and S.N., Methodology: F.L., C.B., C.F.S., D.C., B.R.D.Q., P.Y.P. Investigation: F.L., P.Y.P., F.B., T.P., S.N. Visualisation: F.L. Funding acquisition: S.N. Project administration: S.N. Supervision: F.L., S.N. Writing – original draft: S.N. Writing – review & editing: F.L., C.B., C.F.S., F.B., P.Y.P., T.P., S.N.

## Competing interests

The authors declare no competing interests.
