## [Peer Review File · Nature Communications]

Asymmetric activity of NetrinB controls laterality of the Drosophila brainREVIEWER COMMENTS

Reviewer #1 (Remarks to the Author):

In this well written and comprehensive study, Lapraz et al. have documented and characterised an asymmetric circuit within the fruit fly brain, examining the underlying developmental and genetic processes and relating loss of asymmetry to a behavioural outcome.

Lapraz et al. carefully documented the morphometry, connectivity and frequency of a subset of LALv1A lineage neurons using a specific driver line and through clonal analysis determined that H-neurons located on the left and right sides normally project to and innervate the right AB neuropil. Through a time-course of observations during pupal development, they established that this circuit is initially bilateral and later resolves into a rightward asymmetric form; this process did not seem to involve proliferation or apoptosis of neurons.

The authors established that mutation of *myo1D* (which controls left-right asymmetry of visceral organs) did not affect H-neuron asymmetry, suggesting a brain-dependent secondary mechanism of symmetry-breaking, then performed an RNAi screen, testing genes involved in axonal guidance and midline crossing. Multiple components of the Netrin pathway were identified in the screen, including the two receptors *Unc-5* and *Frazzled*, but not the ligands, *NetA* and *NetB*. Using different driver lines and mutants, the authors concluded that *NetB* is required non-cell autonomously within neuronal subsets of the LALv1A lineage on the right hand side of the brain, but not the H-neurons themselves. In contrast, the receptor, *unc-5* is required by H-neurons on both left and right hand sides. In support of this conclusion, and the timecourse analysis, *unc-5*-GFP fusion protein was observed bilaterally in AB neuropil at 24 hours APF, but unilaterally in the right AB neuropil by 48 hours APF.

A timecourse analysis of *NetB* mutants demonstrated that the initial bilateral circuit stage of H-neurons is formed, but does not resolve into an asymmetric stage. It is worth noting that the initial proportion of "rightward biased" animals in the timecourse is somewhat similar to wildtype, reaching a peak around 40-42 hours APF. The authors also mapped the requirement of *NetB* ligand and *unc-5* receptor over time, using temperature dependent RNAi drivers.

Finally, the authors used a behavioural paradigm to test the functional effect of loss of asymmetry on long term memory formation. They observed a significant loss of performance in *unc-5* RNAi flies 24 hours after a training paradigm involving breaks to consolidate memory, but did not observe overall defects in memory formation after massed or single cycle training with earlier testing.

Questions/comments:

Within the population of flies examined, the authors observed naturally occurring symmetrical projections with an increase in left AB projection volume and decrease in right AB projection volume to match. The authors did not mention if this was related to visceral asymmetry i.e. do the 5% of naturally occurring symmetric flies also have visceral defects? It is true that in 5-10% of zebrafish the habenula have reversed (not randomised) handedness but that reversal is usually concordant with visceral reversal. Consequently, for the fish habenulae, the handedness of the asymmetry is usually coupled with handedness of the viscera whereas asymmetry per se can be uncoupled between body and brain. Terminology distinguishing phenotypes in which asymmetry is absent versus phenotypes in which directional asymmetry (handedness or sometimes termed laterality) is not consistent for papers in this field and the authors should be clear about the terms they use and in making comparisons to asymmetry/handedness phenotypes in other species.

Although the single clone analysis of the spontaneously symmetric flies is consistent with the *NetB* mutant and *unc-5* RNAi data, there is too small a number to draw any firm conclusions.

Whilst true that the *netB* mutant phenotype is broadly similar to the spontaneously symmetric flies, the authors do not mention the fact that the AB neuropil volume is doubled in size.

It is unclear as to how the resolution of the bilateral circuit to an asymmetric one is achieved over time. Is it truly bilateral to begin with, i.e. do both left and right H-neurons equally innervate left and right AB neuropils initially (i.e. both sides projecting contralaterally)? Withdrawal of left H-neuron ipsilateral innervation and right H-neuron contralateral innervation would then be necessary.

The authors suggest that the symmetry breaking event occurs at around 32/34 hrs APF, a point at which there is a clear difference between right and left AB volume (though not noted as significant, Figure 2d). However, it is worth noting that at that timepoint in the analysis, 30% of animals in the sample have already progressed to a “rightward biased” stage. This is similar to the time-course analysis of NetB mutants, where also an apparent point of no return is met at ~40-42 hours APF, suggesting a second decision point.

The authors showed that any genes identified in the 72A10-specific RNAi screen did not show a phenotype when using a pan-glial driver. Did they also test the remaining genes using a pan-glial driver?

Given that there is an apparent hypomorphic effect of unc-5 RNAi with the 72A10 driver (100% flies are overall symmetric, Figure 3b and e, but only ~50% of H-neurons innervate both left and right AB neuropils, Figure 4b), why did the authors choose to use unc-5 RNAi for the behavioural assay? Did they determine that the per-GAL4 driver eliminated this heterogeneity in H-neurons?

There is quite a wide range in left AB and right AB size in symmetric flies, from apparently equal staining on both sides (e.g. per>NetB in Figure 3d) through to very little staining on the left (e.g. Ad2 in Supplementary Figure 5). What was the scoring criteria used?

Frazzled is assumed to also be present in H-neurons on both left and right sides in Figure 6. Could the partial phenotype of frazzled RNAi knockdown (Figure 3e) be explained by asymmetric expression/function of this receptor?

Methods: the description of scoring for period-Gal4 clonal analysis is not at all clear.

Minor points:

Figure 2 panel a: missing scale bars

Figure 2 panels d, e: are these 10 flies selected from the analysis in a and b (n = 20), or was this a separate experiment? If they were selected, what was the criteria? How are bilateral fused neuropil represented in this graph?

Figure 4: was a morphometric analysis of NetB mutants also done (as in figure 2)?

Figure 4 panel f: actual time points (in hours) should be presented on this graph.

Line 37 Main text: dyslexia and schizophrenia are incorrectly capitalised.

Line 752-753 Supplementary Figure 1 Legend: duplication of phrasing.

Supplementary Figure 2: panel titles are unnecessary.

The parallels with left/right asymmetries in fish habenular neuron connectivity are intriguing and the authors may want to further discuss similarities/differences between vertebrate and invertebrate neuronal asymmetries.

Reviewer #2 (Remarks to the Author):

Left-right asymmetry is a fascinating biological phenomenon being essential for correct function of internal organs and brains of vertebrates. Asymmetry is found in invertebrates as well, albeit in a less conspicuous form. Lapraz et al. study the formation of a handful of asymmetric neurons using the power of the *Drosophila melanogaster* model system to investigate the development of brain asymmetry, its genetic underpinning and its function. First, they carefully describe the development of the asymmetry showing that this develops from an initially symmetric situation during a 20 h window in pupal stages. They include extensive quantification e.g. of cell numbers, projections and time course of asymmetry development of the involved H-neurons. After excluding some obvious candidate processes (apoptosis and type 1 myosin), they test 61 candidate axon guidance molecules in an RNAi screen. They identify 10 genes that affect the process 6 of which belonged to the netrin pathway. Thus, they focused on understanding this process. Using mutants, RNAi lines including temporal and clonal control of RNAi mediated knock-down they investigated the timing and the cellular requirement of receptor Unc5 and the ligand NetB. Besides other results, they showed different timing of requirement for receptor and ligand and they found that only NetB showed an asymmetric requirement. Finally, the authors showed that their experimental netrinB-pathway induced loss of asymmetry recapitulated the previously described effect on LTM memory formation - but not on other memory types.

With their work, the authors identify the first genes required for brain laterality in insects including a novel asymmetric requirement for NetrinB. They show that in insects (similar to previous assumptions in vertebrates) the laterality of brain and organs relies on different genetic control. Interestingly, this finding might be transferrable to vertebrates, as asymmetric expression and involvement in laterality issues of netrin components have been described there. Hence, the work is important for fly neurobiologists, for netrin pathway aficionados and for persons considering asymmetries in animals. This work is impressive in its comprehensiveness and the careful analyses using the entire power of the fly model system. Extensive quantifications and statistical analyses make the data very convincing and robust. Finally, the data are presented in a very clear way in both figures and text and very helpful schematics are included. The supplementary data and the methods are extensive.

Congratulations to this work, which was a pleasure to read!

Only minor things that you might consider:

Please write *Drosophila* in italics (convention for species names)

I stumbled over the fact that netrinB mutants seemed to survive to adulthood. To me this was surprising given that it is a gene required for very basic processes of brain formation. Hence, it might be a good idea to specifically mention this for other readers who are just as ignorant about *Drosophila* netrinB as I am.

L 162 “resides into neurons” – did you mean “resides in neurons”?

Gregor Bucher

Reviewer #3 (Remarks to the Author):

In this paper the authors analyse the structural nature of brain asymmetry in the *Drosophila* brain and provide evidence that netrinB signaling is involved in causing the asymmetry. The paper is clearly written and documented, and the findings make a highly significant contribution to our understanding of brain development.

Section 1, lines 40-92

The authors present the basis of asymmetry; using MCFO clones they show that in “asymmetric” (ASYM) flies, a specific population of neurons (H-neurons) projects exclusively to one side of the brain (L-RAB, R-RAB). It is further shown how the H-neuron asymmetry evolves from an initially symmetric pattern during metamorphosis.

Comment:

1.the conclusion from the MCFO analysis should be placed into the context of the recently published Hemibrain connectome, in which all cell types innervating the asymmetric body (AB) and their connectivity are published. The H neurons described by the authors correspond to the ..neurons, which indeed display the L-RAB/R-RAB behavior assigned to the H-neurons by the authors. In addition, the connectome gives a more comprehensive catalog of other AB neurons which do not show an asymmetric branching pattern.

Section 2, lines 94-145

Here the authors focus on the netrinB pathway, which fell out of a screen for genes that affect H-neuron asymmetry. The screening approach is sound and a role of netrinB in asymmetry is convincing. The authors further demonstrate that the netrinB receptor, unc-5, starts to be asymmetrically expressed during early pupal stages

Comment:

2.at the outset the authors state that they look for genes affecting "midline crossing". This is somewhat inaccurate, since according to their developmental analysis, the crossing of H-neurons has already been taking place prior to asymmetry (already larval H-neurons cross the midline, according to Fig.2). That means that the asymmetry is caused by asymmetric formation of side branches of the neurons. Please clarify and discuss.

Section 3, lines 147-

The authors focus on the timepoint and cell group in which netB and its receptor is required. It is also shown that AB asymmetry is required to perform normal in a long term memory assay. Again, the described finding seem conclusive.

Comments:

3.The developmental time course of larval lineages (that during metamorphosis then give rise to adult lineages) have been described in detail in the literature. The lineage in question here that corresponds to LALv1 is called BAMv1. Also the other period-expressing lineages have been described (ALv1 = BALa1, EBA1=DALv2, LALv1A=BAMv1, BALp2=BALp2, SLPp1=DPLI1, DPLm2=DPLm2) (e.g., Pereanu and Hartenstein, 2006; Cardona et al., 2010; Spindler et al., 2009; Spindler and Hartenstein, 2010; Pereanu et al., 2010; Lovick et al., 2013). This literature should be acknowledged.

4.It is hard to see in Fig.S4 that unc-5 is expressed symmetrically at an early stage. To me it looks as if it is not expressed at all in the larva. Please show symmetric expression more convincingly, and/or comment on the early (transcriptional?) mechanism that is responsible for asymmetric expression of this molecule from the beginning.

Reviewer #4 (Remarks to the Author):

Left-Right (LR) asymmetry of the nervous system is an important aspect of healthy brain organization for many functions, such as learning and memory. The authors provide a detailed and decent cellular perspective on LR asymmetry in the nervous system of *Drosophila*. The functional importance of NetB-induced laterality in the fly brain was found for the first time.

Some concerns:

1. An overview of LR asymmetry in the nervous systems of other organisms should be given in the introduction.
2. Dscam serves as one of the receptors of NetB, involved in the axon attraction and axon sister branch segregation, which should be tested.
3. Why NetB in the LALv1A lineage only on the right side is a very important question, one or more of the 28 non-H-neuron types, non-autonomously specifically on the right side? It hasn't been directly addressed yet in the manuscript..

We wish to thank the Reviewers for their strong support and suggestions, which helped improve our manuscript both in clarity and content. For readability, all changes made to the revised text have been highlighted.

Please note that in addition to our response to the reviewer's points (see below), we have added new behavioural data which we completed since submission. In collaboration with the Besse lab, we performed additional memory tests using a complementary paradigm, courtship suppression behaviour. The new data nicely confirm our original data (which used an aversive associative olfactory paradigm; fig6b), by showing that during courtship conditioning, *unc-5* RNAi flies also show defects in long-term memory without affecting short-term memory. We believe the addition of these data will help reinforce and extend our findings. The results have been added to fig6 as new panels (fig6c) and described in the revised text (lines 213-219).

REVIEWER COMMENTS

Reviewer #1 (Remarks to the Author):

In this well written and comprehensive study, Lapraz et al. have documented and characterised an asymmetric circuit within the fruit fly brain, examining the underlying developmental and genetic processes and relating loss of asymmetry to a behavioural outcome.

Lapraz et al. carefully documented the morphometry, connectivity and frequency of a subset of LALv1A lineage neurons using a specific driver line and through clonal analysis determined that H-neurons located on the left and right sides normally project to and innervate the right AB neuropil. Through a time-course of observations during pupal development, they established that this circuit is initially bilateral and later resolves into a rightward asymmetric form; this process did not seem to involve proliferation or apoptosis of neurons.

The authors established that mutation of *myo1D* (which controls left-right asymmetry of visceral organs) did not affect H-neuron asymmetry, suggesting a brain-dependent secondary mechanism of symmetry-breaking, then performed an RNAi screen, testing genes involved in axonal guidance and midline crossing. Multiple components of the Netrin pathway were identified in the screen, including the two receptors *Unc-5* and *Frazzled*, but not the ligands, *NetA* and *NetB*. Using different driver lines and mutants, the authors concluded that *NetB* is required non-cell autonomously within neuronal subsets of the LALv1A lineage on the right hand side of the brain, but not the H-neurons themselves. In contrast, the receptor, *unc-5* is required by H-neurons on both left and right hand sides. In support of this conclusion, and the timecourse analysis, *unc-5*-GFP fusion protein was observed bilaterally in AB neuropil at 24 hours APF, but unilaterally in the right AB neuropil by 48 hours APF.

A timecourse analysis of *NetB* mutants demonstrated that the initial bilateral circuit stage of H-neurons is formed, but does not resolve into an asymmetric stage. It is worth noting that the initial proportion of "rightward biased" animals in the timecourse is somewhat similar to wildtype, reaching a peak around 40-42 hours APF. The authors also mapped the requirement of *NetB* ligand and *unc-5* receptor over time, using temperature dependent RNAi drivers.

Finally, the authors used a behavioural paradigm to test the functional effect of loss of asymmetry on long term memory formation. They observed a significant loss of performance in *unc-5* RNAi flies 24 hours after a training paradigm involving breaks to

consolidate memory, but did not observe overall defects in memory formation after massed or single cycle training with earlier testing.

Questions/comments:

1. Within the population of flies examined, the authors observed naturally occurring symmetrical projections with an increase in left AB projection volume and decrease in right AB projection volume to match. The authors did not mention if this was related to visceral asymmetry i.e. do the 5% of naturally occurring symmetric flies also have visceral defects? It is true that in 5-10% of zebrafish the habenula have reversed (not randomised) handedness but that reversal is usually concordant with visceral reversal. Consequently, for the fish habenulae, the handedness of the asymmetry is usually coupled with handedness of the viscera whereas asymmetry per se can be uncoupled between body and brain.

Re: Thank you for this interesting point. We have not checked the concordance directly, because LR asymmetry of the viscera (e.g., genitalia, hindgut) is extremely robust in wild type (and most mutant) flies. Based on phenotypic analysis over the years and published work from our and other labs (e.g., Hayashi et al., 2001 DOI: 10.1046/j.1440-169x.2001.00574.x; Hozumi et al., 2006 doi: 10.1038/nature04625; Spéder et al. 2006 doi: 10.1038/nature04623; Gonzalez-Morales et al., 2015 doi: 10.1016/j.devcel.2015.04.026), the frequency of spontaneous defects in visceral asymmetry is found to vary from 0%-to-0.5% (compared to about 5%-7.5% for H-neurons; this study and Pascual et al., 2004), indicating that brain and viscera asymmetry are uncoupled in *Drosophila*, which is also supported by our results showing that *myo1D* mutants do not affect brain asymmetry (figS2b).

The usually concordant zebrafish mutants may be linked to the fact that nodal in fish is involved in both visceral and brain asymmetry. However, it also seems that wildtype and some particular mutants (e.g., *cycb229*, *mom*, etc..) can show moderate to strong discordance between brain vs. heart or gut (*Bisgrove et al., 2000 DOI 10.1242/dev.127.16.3567*), suggesting that concordance may vary depending on the species and genetic background.

2. Terminology distinguishing phenotypes in which asymmetry is absent versus phenotypes in which directional asymmetry (handedness or sometimes termed laterality) is not consistent for papers in this field and the authors should be clear about the terms they use and in making comparisons to asymmetry/handedness phenotypes in other species.

Re: we agree with the Reviewer that terminology is not always consistent across the literature. In our work, we use LR asymmetry, laterality (and chirality in the case of visceral organs undergoing coiling) as equivalent terms to describe structural LR asymmetry (involving lateralised positioning and/or morphology of organs). We use the term handedness only to describe the hand (limb) preference (as a behaviour), like is mentioned in our text (lines 35 and 245).

3. Although the single clone analysis of the spontaneously symmetric flies is consistent with the *NetB* mutant and *unc-5* RNAi data, there is too small a number to draw any firm conclusions.

Re: we agree with the reviewer that a larger number would help drawing a firmer conclusion, which is why we mention that what makes SYM and *netB/unc-5* similar is the presence of bilateral neurons, a category not seen in ASYM flies (line 144). Increasing the number of single clones for the SYM condition would represent a major experimental

effort, since we had to analyse 1088 individual MCFO brains to get 6 SYM brains with a single clone. This is due to the fact that SYM brains are rare (5%) combined with the low frequency of single cell clone generation using the MCFO technique. As mentioned above, despite the relatively low number, the phenotype distributes into discrete categories (e.g., presence/absence of bilateral neurons) rather than being continuous (like for example, weight in a population), making comparisons more significant. In order to strengthen our conclusion, we have performed statistical analysis on the data using the Fisher test, showing that the *netB/unc5* mutants are indeed not significantly different from SYM flies. We have added this analysis to a new figS7, thank you for raising the point which helped make a stronger conclusion.

4. Whilst true that the *netB* mutant phenotype is broadly similar to the spontaneously symmetric flies, the authors do not mention the fact that the AB neuropil volume is doubled in size.

Re: thank you for the insightful comment. We think the increased neuropil volume observed in *netB* null mutants is unspecific, as the more specific silencing in H-neurons (*72A10>unc-5 RNAi* flies) does not show such a change (figS1a). As suggested, we are now mentioning the increased AB neuropil size, as follows (lines 144-146): 'We noted that the AB volume of *netB* mutants is increased as compared to SYM flies (figS1a), which is not linked to H-neuron number, and may be related to additional, unspecific effects of the mutation.'

5. It is unclear as to how the resolution of the bilateral circuit to an asymmetric one is achieved over time. Is it truly bilateral to begin with, i.e. do both left and right H-neurons equally innervate left and right AB neuropils initially (i.e. both sides projecting contralaterally)? Withdrawal of left H-neuron ipsilateral innervation and right H-neuron contralateral innervation would then be necessary.

Re: labelling of H-neurons at pupal stage shows fibers connecting LAB and RAB (please see fig2a and fig4c; and close-up shown below), indicating the presence of contralateral projections from 28hrs APF. Hence, we fully agree with the reviewer that both left (ipsilateral) and right (contralateral) innervation have to be withdrawn to make the mature ASYM phenotype.

Legend to rebuttal figure1:

Labelling of ABs during pupal development (from 28hrs to 44hrs APF), showing fibers crossing the midline and linking both ABs (marked with arrowheads). mCherry (32hrs, 42hrs, 44hrs) or GFP (28hrs, 38hrs, 40hrs) were expressed in H-neurons (using 72A10-lexA) to label projections.

6. The authors suggest that the symmetry breaking event occurs at around 32/34 hrs APF, a point at which there is a clear difference between right and left AB volume (though not noted as significant, Figure 2d). However, it is worth noting that at that timepoint in the analysis, 30% of animals in the sample have already progressed to a "rightward biased" stage. This is similar to the time-course analysis of *NetB* mutants, where also an apparent point of no return is met at ~40-42 hours APF, suggesting a second decision point.

Re: For simplicity, we defined the symmetry-breaking point as the time when the first right-only patterns are observed (32-34hrs). We fully agree with the reviewer that the curve is more complex and suggest the presence of two phases: one *netB*-independent phase (28-42hrs) during which 'rightward biased' brains increase and reach a plateau or maximum (grey curves in fig2b and fig4d); and second, a *netB*-dependent phase during which 'rightward biased' brains decrease (42-48hrs) for the benefit of right-only or bilateral circuits, as observed in wildtype and *netB* mutants, respectively; We have added a sentence to better reflect this complexity, as follows (lines 155-159): 'Comparison of wildtype (Fig. 2b) and *netB* mutant (Fig. 4d) curves suggests the presence of two phases: one *netB*-independent phase (28-42hrs) during which bilateral 'rightward biased' brains increase and reach a plateau or maximum (grey curves in fig2b and fig4d); and second, a *netB*-dependent phase during which bilateral 'rightward biased' brains decrease

(40-42hrs) for the benefit of right-only or bilateral circuits, as is observed in wildtype and *netB* mutants, respectively.'

7. The authors showed that any genes identified in the 72A10-specific RNAi screen did not show a phenotype when using a pan-glial driver. Did they also test the remaining genes using a pan-glial driver?

Re: only the positive hits have been tested for a potential glial phenotype. We made the text clearer on this point, as follows (lines 123-125): "Driving RNAi against the 10 positive hits using a pan-glial driver (*gcm-Gal4*) did not show a phenotype, indicating that the laterality defects are neuronal-specific (Fig. S2c). »

8. Given that there is an apparent hypomorphic effect of *unc-5* RNAi with the 72A10 driver (100% flies are overall symmetric, Figure 3b and e, but only ~50% of H-neurons innervate both left and right AB neuropils, Figure 4b), why did the authors choose to use *unc-5* RNAi for the behavioural assay? Did they determine that the *per-GAL4* driver eliminated this heterogeneity in H-neurons?

Re: the *unc-5* RNAi condition was used because it gives the strongest phenotype (please see fig3b,d,e). For behavioural studies, we had to choose a condition leading to the strongest possible and specific bilateral phenotype, at 18°C and without *dicer*, which is required to perform the olfactory behavioural tests. Only the *period>unc-5* RNAi condition fit these constraints, leading to a strong bilateral phenotype, as illustrated in the rebuttal figure below comparing *per-Gal4* and 72A10-Gal4 drivers with flies raised at 18°C, without *dicer*.

The 72A10-Gal4 driver was chosen for single neuron analysis (fig4b) for technical reasons, i.e., to reduce the complexity of clones by restricting single clones to the functional 72A10 domain, instead of the whole LALv1A domain and the other 8 identified clonal domains generated with *per-Gal4* (please see fig5a,b). The analysis of volumetry revealed the hypomorphic nature of this condition, thus making it possible to correlate the proportion of bilateral neurons to AB's volume (figS1a).

Legend to rebuttal figure2:

Phenotype of per>unc-5 RNAi (top panels) and 72A10>unc-5 RNAi (bottom panels) brains labelled with anti-Fas2 antibody to reveal the ABs. Flies were raised at 18°C with no dicer2 (conditions required for olfactory behavioural tests). Data show that per-Gal4 induces 100% bilateral brains (n=10), while only 44% (n=9) are observed when unc-5 RNAi is driven by 72A10-Gal4. The per>unc-5 RNAi images are the same as the ones shown in figS5 (note: the original figS5 was in colour and is now shown in black and white for better contrast; also, arrowheads pointing to ABs have been added for clarity). Arrowheads show labelling in left (red) and right (blue) ABs.

9. There is quite a wide range in left AB and right AB size in symmetric flies, from apparently equal staining on both sides (e.g. per>NetB in Figure 3d) through to very little staining on the left (e.g. Ad2 in Supplementary Figure 5). What was the scoring criteria used?

Re: for scoring, we used a wide inclusive criterium in which flies were scored as being SYM when they showed projections in the LAB regardless of their volume. Since, as also pointed out by the reviewer, we noticed some variability in LAB volume, we provide volumetric analysis of AB for 4 conditions in figS1a (ASYM, SYM, NetB mutant and 72A10>unc-5 RNAi). The latter two conditions have been included purposely because they induced phenotypes with large and small LAB volumes, respectively.

Of note, we observe fluctuating volumes in wildtype flies (see also Wolff et al., 2018), which we think is intrinsic to the system as a mean to generate diversity in the population (as discussed in the text; lines 246-248)

10. Frazzled is assumed to also be present in H-neurons on both left and right sides in Figure 6. Could the partial phenotype of frazzled RNAi knockdown (Figure 3e) be explained by asymmetric expression/function of this receptor?

Re: we do not think the partial phenotype of fra RNAi knockdown is indicative of an asymmetric function of this receptor (for example, netB which has a unilateral function does not show a partial phenotype), instead different potential causes like RNAi efficiency, timing, driver strength, etc.. may lead to the partial effect. However, we do not exclude the possibility that fra could also have an asymmetrical function. We tried addressing this question directly, however faced technical limitations; indeed, because the phenotype is partial, we could not use unilateral clonal analysis like with NetB mutant or unc-5 RNAi, since a prerequisite for this experiment is to have a robust phenotype.

11. Methods: the description of scoring for period-Gal4 clonal analysis is not at all clear.

Re: the description of per-Gal4 clonal analysis has been clarified (lines 370-386).

Minor points: thank you for raising these points, which helped make the data and text clearer.

Figure 2 panel a: missing scale bars

Re: scale bars have been added to the panel 2a, and 4a for which it was also missing.

Figure 2 panels d, e: are these 10 flies selected from the analysis in a and b (n = 20), or was this a separate experiment? If they were selected, what was the criteria? How are bilateral fused neuropil represented in this graph?

Re: for panels a-e, the same samples have been used; in b, each time point is the sum of 2 time points (n=10 each) from curves d,e, except for 48hrs (n=10)(as indicated in legend to fig2b; please note that this information was mistakenly provided to fig2a and has now been moved; lines 811-813).

The volume of bilateral fused flies has been determined by drawing the midline (please see fig2a, 30hrs) and the volume calculated from each side.

Figure 4: was a morphometric analysis of NetB mutants also done (as in figure 2)?

Re: we have performed morphometric analysis for netB mutants. The results have been added to revised fig4 (panels f,g).

Figure 4 panel f: actual time points (in hours) should be presented on this graph.

Re: times points have been added to the graph.

Line 37 Main text: dyslexia and schizophrenia are incorrectly capitalised.

Re: we have corrected the text by removing the capitals.

Line 752-753 Supplementary Figure 1 Legend: duplication of phrasing.

Re: the duplicated text has been removed.

Supplementary Figure 2: panel titles are unnecessary.

Re: the panel titles have been removed

The parallels with left/right asymmetries in fish habenular neuron connectivity are intriguing and the authors may want to further discuss similarities/differences between vertebrate and invertebrate neuronal asymmetries.

Re: we have highlighted some analogies between H-neurons and habenula in the discussion (lines 230-234)

Reviewer #2 (Remarks to the Author):

Left-right asymmetry is a fascinating biological phenomenon being essential for correct function of internal organs and brains of vertebrates. Asymmetry is found in invertebrates as well, albeit in a less conspicuous form. Lapraz et al. study the formation of a handful of asymmetric neurons using the power of the *Drosophila melanogaster* model system to investigate the development of brain asymmetry, its genetic underpinning and its function. First, they carefully describe the development of the asymmetry showing that this develops from an initially symmetric situation during a 20 h window in pupal stages. They include extensive quantification e.g. of cell numbers, projections and time course of asymmetry development of the involved H-neurons. After excluding some obvious candidate processes (apoptosis and type 1 myosin), they test 61 candidate axon guidance molecules in an RNAi screen. They identify 10 genes that affect the process 6 of which belonged to the netrin pathway. Thus, they focused on understanding this process. Using mutants, RNAi lines including temporal and clonal control of RNAi mediated knock-down they investigated the timing and the cellular requirement of receptor Unc5 and the ligand NetB. Besides other results, they showed different timing of requirement for receptor and ligand and they found that only NetB showed an asymmetric requirement. Finally, the authors showed that their experimental netrinB-pathway induced loss of asymmetry recapitulated the previously described effect on LTM memory formation - but not on other memory types. With their work, the authors identify the first genes required for brain laterality in insects including a novel asymmetric requirement for NetrinB. They show that in insects (similar to previous assumptions in vertebrates) the laterality of brain and organs relies on

different genetic control. Interestingly, this finding might be transferrable to vertebrates, as asymmetric expression and involvement in laterality issues of netrin components have been described there. Hence, the work is important for fly neurobiologists, for netrin pathway aficionados and for persons considering asymmetries in animals. This work is impressive in its comprehensiveness and the careful analyses using the entire power of the fly model system. Extensive quantifications and statistical analyses make the data very convincing and robust. Finally, the data are presented in a very clear way in both figures and text and very helpful schematics are included. The supplementary data and the methods are extensive.

Congratulations to this work, which was a pleasure to read!

Only minor things that you might consider: thank you for the strong support and for raising these points, which helped make the data and text clearer.

Please write *Drosophila* in italics (convention for species names)

Re: Drosophila has been written in italics

I stumbled over the fact that netrinB mutants seemed to survive to adulthood. To me this was surprising given that it is a gene required for very basic processes of brain formation. Hence, it might be a good idea to specifically mention this for other readers who are just as ignorant about *Drosophila* netrinB as I am.

Re: ‘despite being viable’ has been added to the text (line 137). The fact that netB mutants are viable may be due to partial redundancy with netA in some processes.

L 162 “resides into neurons” – did you mean “resides in neurons”?

Re: yes indeed, the text has been corrected.

Gregor Bucher

Reviewer #3 (Remarks to the Author):

In this paper the authors analyse the structural nature of brain asymmetry in the *Drosophila* brain and provide evidence that netrinB signaling is involved in causing the asymmetry. The paper is clearly written and documented, and the findings make a highly significant contribution to our understanding of brain development.

Section 1, lines 40-92

The authors present the basis of asymmetry; using MCFO clones they show that in “asymmetric” (ASYM) flies, a specific population of neurons (H-neurons) projects exclusively to one side of the brain (L-RAB, R-RAB). It is further shown how the H-neuron asymmetry evolves from an initially symmetric pattern during metamorphosis.

Comment:

1. the conclusion from the MCFO analysis should be placed into the context of the recently published Hemibrain connectome, in which all cell types innervating the asymmetric body (AB) and their connectivity are published. The H neurons described by the authors correspond to the ..neurons, which indeed display the L-RAB/R-RAB behavior assigned to

the H-neurons by the authors. In addition, the connectome gives a more comprehensive catalog of other AB neurons which do not show an asymmetric branching pattern.

Re: thank you for suggesting linking the connectome to our study. We have analysed the hemibrain data, with results indicating that the female brain used for the connectome is in fact an ASYM brain, with 8 L-RAB H-neurons on the left side and 9 R-RAB on the right side, well confirming our results (figS1b). The analysis is presented in a new supplementary figure (figS6), showing the individual morphology, annotation and listing of left and right H-neurons. We have added a sentence to put our results in the context of the connectome, as follows (lines 97-99): "*We analyzed the recently published hemibrain connectome indicating that the female used was an ASYM fly, with 8 L-RAB neurons on the left and 9 R-RAB neurons on the right (Fig. S6), in agreement with our data (Fig. S1b).*" A dedicated section has been added to the Methods (lines 448-452).

Section 2, lines 94-145

Here the authors focus on the netrinB pathway, which fell out of a screen for genes that affect H-neuron asymmetry. The screening approach is sound and a role of netrinB in asymmetry is convincing. The authors further demonstrate that the netrinB receptor, unc-5, starts to be asymmetrically expressed during early pupal stages

Comment:

2. at the outset the authors state that they look for genes affecting "midline crossing". This is somewhat inaccurate, since according to their developmental analysis, the crossing of H-neurons has already been taking place prior to asymmetry (already larval H-neurons cross the midline, according to Fig.2). That means that the asymmetry is caused by asymmetric formation of side branches of the neurons. Please clarify and discuss.

Re: The rationale for selecting 'midline crossing' genes came from the fact that in adults, we guessed that left H-neurons projected in the right AB and hence have to cross the midline, which was later confirmed by our MCFO data on wildtype SYM and ASYM flies. Also, our screen was performed before we analysed the H-neurons during pupal development. Since H-neurons have to cross the midline, we believe the rationale of our screen still holds true.

Please note that H-neurons are not detected at larval stage, since using the 72A10-lexA driver allows to detect H-neurons from 28hrs after pupal formation (APF) onward.

As discussed above (please see Reviewer 1 point 5) and in the text (lines 196-199), following an initial symmetrical pattern (default state), netB is likely involved in attracting and maintaining right-only projections.

Section 3, lines 147-

The authors focus on the timepoint and cell group in which netB and its receptor is required. It is also shown that AB asymmetry is required to perform normal in a long term memory assay. Again, the described finding seem conclusive.

Comments:

3. The developmental time course of larval lineages (that during metamorphosis then give rise to adult lineages) have been described in detail in the literature. The lineage in question here that corresponds to LALv1 is called BAMv1. Also the other period-expressing lineages have been described (ALv1 = BALa1, EBa1=DALv2, LALv1A=BAMv1, BALp2=BALp2, SLPp1=DPLI1, DPLm2=DPLm2) (e.g., Peraanu and Hartenstein, 2006; Cardona et al., 2010; Spindler et al., 2009; Spindler and Hartenstein, 2010; Peraanu et al., 2010; Lovick et al., 2013). This literature should be acknowledged.

Re: We are sorry if the information was not clear enough. All synonyms were already cited in the Methods section (lines 353-360), except for SLPpl1/DPLI1 (this has now been corrected, thank you; line 360), as well as some of the listed references. Other references were not cited because they referred to larval stage instead of the adult stage in which we scored and identified the labelled clones. As suggested by the reviewer, we added the following references: *Pereanu and Hartenstein, 2006/10.1523/JNEUROSCI.4708-05.2006; Spindler and Hartenstein, 2010/10.1007/s00427-010-0323-7; Pereanu et al., 2010/10.1002/cne.22376 ; Lovick et al., 2013/10.1016/j.ydbio.2013.07.008*

4. It is hard to see in Fig.S4 that unc-5 is expressed symmetrically at an early stage. To me it looks as if it is not expressed at all in the larva. Please show symmetric expression more convincingly, and/or comment on the early (transcriptional?) mechanism that is responsible for asymmetric expression of this molecule from the beginning.

Re: we are sorry if our initial figure was unclear. We do not mention any expression at larval stages and indeed agree with the reviewer that we do not see any signal at L2 and L3 stages (figS4), which is consistent with the unc-5 functional window (fig4f). To avoid any misunderstanding, we now mention in the text that we do not see labelling at larval stages. (line 927).

Labelling suggests that unc-5 expression mirrors the dynamic pattern of H-neuron's projections, being present in the 2 ABs in early stages and only in the right one when the pattern fully matures. By saying symmetrical expression, we meant in both ABs, the slight difference between left and right being a consequence of their difference in volume already at this early stage (please see fig2d). We have changed the text to make it clearer (lines 925-926).

Reviewer #4 (Remarks to the Author):

Left-Right (LR) asymmetry of the nervous system is an important aspect of healthy brain organization for many functions, such as learning and memory. The authors provide a detailed and decent cellular perspective on LR asymmetry in the nervous system of *Drosophila*. The functional importance of NetB-induced laterality in the fly brain was found for the first time.

Some concerns:

1. An overview of LR asymmetry in the nervous systems of other organisms should be given in the introduction.

Re: thank you for suggesting this point, which helped putting our study in a broader context. We have expanded our introduction and cite some representative literature from other model systems, including zebrafish (lines 41-57).

2. Dscam serves as one of the receptors of NetB, involved in the axon attraction and axon sister branch segregation, which should be tested..

Re: Dscam was tested in our original screen (using 2 different RNAi lines; please see Table S1). Our results show that Dscam RNAi does not give a significant phenotype and hence does not seem to play a role in the process.

3. Why NetB in the LALv1A lineage only on the right side is a very important question, one or more of the 28 non-H-neuron types, non-autonomously specifically on the right side? It hasn't been directly addressed yet in the manuscript..

Re: we agree with the reviewer that identifying the right LAV1A lineage neurons that are required for NetB function is an important question, yet difficult to address at this stage. This would require identifying appropriate drivers expressed specifically in a small subset of neurons at the appropriate developmental stage. We could not identify such drivers despite our efforts (we screened several dozens of GAL4, split-Gal4 combinations, lexA drivers unsuccessfully). As an alternative approach, we screened available NetB reporter lines, but the expression pattern was diffuse and broad, and lines generated a phenotype on their own making them unsuitable to study H-neurons (please see figS2d and text, lines 167-170). Hence, identifying these neurons will require substantial efforts (including developing new tools) which we plan to perform in future work.

REVIEWERS' COMMENTS

Reviewer #1 (Remarks to the Author):

The authors have done a good job of addressing reviewer comments and we are happy to recommend the revised manuscript for publication without further revision.

Reviewer #3 (Remarks to the Author):

The authors have done a careful and comprehensive job in revising their manuscripts. I have no further objections towards publishing the paper.

Reviewer #4 (Remarks to the Author):

Why NetB in the LALv1A lineage only on the right side is a very important question, one or more of the 28 non-H-neuron types, non-autonomously specifically on the right side? It hasn't been directly addressed yet in the manuscript. Identifying the neurons in charge of NetB function will require more efforts. Single-cell sequencing data in drosophila has been extensively analyzed, which may help to trace the clue of Left-Right (LR) asymmetry of the nervous system. I expected the authors could make a systematic and comprehensive work on this interesting topic in the future.

REVIEWERS' COMMENTS

Reviewer #1 (Remarks to the Author):

The authors have done a good job of addressing reviewer comments and we are happy to recommend the revised manuscript for publication without further revision.

Re: We wish to thank the reviewer for the strong support.

Reviewer #3 (Remarks to the Author):

The authors have done a careful and comprehensive job in revising their manuscripts. I have no further objections towards publishing the paper.

Re: We wish to thank the reviewer for the strong support.

Reviewer #4 (Remarks to the Author):

Why NetB in the LALv1A lineage only on the right side is a very important question, one or more of the 28 non-H-neuron types, non-autonomously specifically on the right side? It hasn't been directly addressed yet in the manuscript. Identifying the neurons in charge of NetB function will require more efforts. Single-cell sequencing data in drosophila has been extensively analyzed, which may help to trace the clue of Left-Right (LR) asymmetry of the nervous system. I expected the authors could make a systematic and comprehensive work on this interesting topic in the future.

Re: We wish to thank the reviewer for the strong support and suggestion to perform single-cell sequencing to identify NetB neurons from the LALv1 lineage. Doing this is in fact one of the approaches we will use in future work to address this important question and more generally to bring single-cell resolution to the problem of brain laterality.